

# Curriculum Vitae of the LOTOS-EUROS (v2.0) chemistry transport model

Astrid M. M. Manders[1], Peter J.H. Builtjes[1,4], Lyana Curier[1], Hugo A.C. Denier van der Gon[1], Carlijn Hendriks[1], Sander Jonkers[1], Richard Kranenburg[1], Jeroen Kuenen[1], Arjo J. Segers[1], Renske M.A. Timmermans[1], Antoon Visschedijk[1], Roy J. Wichink Kruit[2], W. Addo J. van Pul[2], Ferd J. Sauter[2], Eric van der Swaluw[2], Daan P.J. Swart[2], John Douros[3], Henk Eskes[3], Erik van Meijgaard[3], Bert van Ulft[3], Peter van Velthoven[3], Sabine Banzhaf[4], Andrea Mues[4], Rainer Stern[4], Guangliang Fu[5], Sha Lu[5], Arnold Heemink[5], Nils van Velzen[6], Martijn Schaap[1,4]

[1] TNO, PO Box 80015, 3508TA Utrecht, The Netherlands,
[2] RIVM, Antonie van Leeuwenhoeklaan 9, 3721 MA Bilthoven, The Netherlands
[3] KNMI, PO Postbus 201, 3730 AE De Bilt, The Netherlands
[4] Institute of Meteorology, Freie Universität, Berlin Carl-Heinrich-Becker-Weg 6-10, 12165, Berlin, Germany
[5] TU Delft, P.O. Box 5031 2600 GA Delft, The Netherlands
[6] VORrtech, Martinus Nijhofflaan 2, 2624 ES Delft, The Netherlands

*Correspondence to*: A. M. M. Manders (astrid.manders@tno.nl)



**Abstract**

The development and application of chemistry transport models has a long tradition. Within the Netherlands the LOTOS-EUROS model has been developed by a consortium of institutes, after combination of its independently developed predecessors in 2005. Recently, version 2.0 of the model was released as an open source version. This paper presents the
curriculum vitae of the model system, describing the model's history, model philosophy, basic features, a validation with EMEP stations for the new benchmark year 2012, and presents cases with the model's most recent and key developments. By setting the model developments in context and providing an outlook for directions for further development, the paper goes beyond the common model description.

With an origin in ozone and sulphur modelling for the models LOTOS and EUROS, the application areas were gradually extended with POPs, reactive nitrogen and primary and secondary particulate matter. After the combination of the models to LOTOS-EUROS in 2005, the model was further developed to include new source parametrizations (e.g. road resuspension, desert dust, wildfires), applied for operational smog forecasts in the Netherlands and Europe, and has been used for emission scenarios, source apportionment and long-term hindcast and climate change scenarios. LOTOS-EUROS has been a front-
runner in data assimilation of ground-based and satellite observations and has participated in many model intercomparison studies. The model is no longer confined to applications over Europe but is also applied to other regions of the world, e.g. China. Also the increasing interaction with emission experts has contributed to the improvement of the model's performance. The philosophy for model development has always been to use knowledge that is state of the art and proven, to keep good balance in the level of detail of process description and accuracy of input and output, and to keep a good track on the effect
of model changes using benchmarking and validation. The performance of v2.0 with respect to EMEP observations is good, with spatial correlations around 0.8 or higher for concentrations and wet deposition. Temporal correlations are around 0.5 or higher. Recent innovative applications include source apportionment and data assimilation, particle number modelling, energy transition scenarios including corresponding land use changes as well as Saharan dust forecasting. Future developments would enable more flexibility with respect to model horizontal and vertical resolution and further detailing of model input data. This
includes use of different sources of land use characterization (roughness length and vegetation), detailing of emissions in space and time, and efficient coupling to meteorology from different meteorological models.



# 1    Introduction

The most pressing environmental challenges relate to the composition of the atmosphere. Air pollution, climate change and ecosystem degradation have wide-ranging effects on human well-being as well as biodiversity and affect sustainable growth in general. Air pollution has been recognized as a harm for public health and the environment since the 1950's, with the recognition of elevated tropospheric ozone levels in Los Angeles (Haagen-Smit, 1952). The impact of acid deposition was recognized in Europe in the 1950's (Chamberlain, 1953). Whereas air quality was originally regarded as an urban problem, large scale acidification of soils and surface water as well as summertime ozone episodes made it clear that air quality was a transboundary problem that needed to be solved at the international level (Eliassen, 1978). Based on this consideration the Convention on Long Range Transboundary Air Pollution (CLRTAP) was established in 1979. Although emission reduction strategies have been successful for a number of pollutants, air pollution is still an issue. It largely contributes to the burden of lung cancer and respiratory and cardiovascular diseases, which are associated with a considerable loss of life expectancy (EEA, 2016).

In the early 1970s the first models for air pollution were developed in the US, mainly aimed at studying episodic photochemistry of ozone (e.g. Reynolds et al, 1973).  Simultaneously, models aimed at analyzing acid deposition were developed in Europe (Rohde, 1972. In the beginning, in the US 3-D Eulerian grid models were preferred while  trajectory models  were favored in Europe.  The difference was partly motivated by  the focus of the models: ozone in the US and deposition in Europe, but also the background of the scientists played a role; atmospheric chemists in the US and meteorologists in Europe.  To underpin cost effective mitigation strategies for air pollution, chemistry transport models were further developed and applied in Europe under the LRTAP convention and within the member states.  During the subsequent decades the scope of application of chemistry transport models has increased enormously to study acid rain (e.g. Eliassen and Saltbones, 1983), particulate matter (e.g. Mebust et al 2003; Schaap et al., 2004b), reactive nitrogen (e.g. Derwent et al 1989), persistent pollutants (e.g. Pekar et al 1998) and mercury (Ryaboshabko, 2002).

Besides fundamental research , chemistry transport models are nowadays used for operational chemical weather forecasting (e.g. Marecal et al., 2015), air quality reanalyses on annual to decadal time scales (e.g. Andersson et al., 2007, Banzhaf et al., 2015), exploring mitigation measures either by direct comparison of scenario simulations (Thunis et al., 2008, 2010) or indirectly by providing underlying material for assessment models like GAINS (Amman et al.,  2011 ), climate change through coupling with climate models (e.g. Jacob and Winner, 2009) and modelling of feedbacks between meteorology and aerosols by on-line coupled numerical weather and chemistry models (e.g. Baklanov et al., 2014) as well as designing monitoring strategies using in-situ observations or new satellite instruments (e.g. Timmermans et al., 2015). Nowadays, a large number of CTMs exist with a few widely used open source systems such as EMEP (Simpson et al., 2012), CHIMERE (Menut et al., 2013a), WRF-CHEM (Fast et al 2006, Grell et al  2004), CMAQ (Byun and Schere, 2006) and CAMx (Environ, 2014).



In contrast to the US, in Europe a variety of air quality models was developed with relatively small user communities for each model since countries invested in their own model. Model intercomparison exercises have contributed to the acceptance of these models and determined the robustness of single model results for policy support purposes (starting with Hass et al 1997 and continuing with TFMM-EURODELTA, Collette et al 2017).

Within the Netherlands the LOTOS-EUROS model has been developed by a consortium of institutes. The model system originates from a merger of two model systems (LOTOS and EUROS) developed individually since the eighties at TNO and RIVM. After integration in 2005 an overview paper was published (Schaap et al., 2008). During the last ten years there have been numerous changes involving new or revised parameterisations, additional functionalities and application areas. In

addition, the LOTOS-EUROS system was released in an open source version in 2016. The reason for open source is to increase the number of users and developers, which would make the basis for the model more solid and would enhance further model development.

In this paper we present LOTOS-EUROS v2.0 and its Curriculum Vitae. Other models have published their model description

(e.g. Simpson et al. (2012) for EMEP and Menut et al. (2013a) for CHIMERE) in peer-reviewed journals. Since the LOTOS-EUROS reference guide is already available at the LOTOS-EUROS website we do not feel the need for a peer-reviewed version. However, a kind of model CV reflecting the long-term model evolution, model portfolio as well as development and benchmarking strategy has not been published before. Such an overview gives a broader perspective on the model philosophy and research directions and is complementary to the regular documentation. First, the model history is presented, relating key

developments to societal questions, new scientific knowledge and technical possibilities. Second, the model development and benchmarking strategy is presented. Next, an overview of the model version 2.0 is provided and complemented with the results of the new internal benchmark test. The model portfolio is then sketched with illustrations of special features like source apportionment and data assimilation. Finally possibilities and motivations for further model improvement will be outlined.



## 2    History

### 2.1    Origin of LOTOS-EUROS

LOTOS-EUROS started as two separate models. Since the late 1970s the Dutch institutes RIVM and TNO have independently
developed their Eulerian models to calculate the dispersion and chemical transformation of air pollutants in the lower
troposphere over Europe.

The LOng Term Ozone Simulation (LOTOS) model originates from the US Urban Airshed Model (UAM). In the early 1970's,
Steven Reynolds and colleagues in the group of John Seinfeld at Caltech and later at the private firm Systems Applications
Incorporated (SAI) made the pioneering attempts at modelling photochemical air quality (Reynolds et al, 1973). These efforts
resulted in the UAM model, a local Eulerian grid air quality model focused on ozone in episodic situations in urbanized areas.
It was firstly designed to investigate ozone formation over Los Angeles (US).  Los Angeles showed the highest peak levels of
ozone concentrations that were a major concern in the US,  and UAM was used for emission scenario studies.  In a cooperation
between SAI and TNO, the UAM was modified for application over the Netherlands and its surroundings (Builtjes et al, 1980,
Builtjes et al, 1982). In the beginning of the 1980's, TNO and SAI started cooperation with the FU Berlin (Freie Universität
Berlin, Institute of Meteorology) to apply UAM for parts of Germany. In the 1980's the awareness increased that next to
episodic ozone, also more long term values were of importance. In the US, SAI extended the UAM to cover larger areas and
longer periods, which was partly possible due to the increase in computer resources. The new model  was subsequently called
RTM (Regional Transport Model). RTM is the direct predecessor of LOTOS (Builtjes, 1992).  LOTOS contained special
features of the original UAM/RTM. It is the vertical structure with a  time-varying mixing layer and two reservoir layers which
makes the now called LOTOS model unique in its existence. It was later extended with aerosol components (Schaap et al ,
2004a, 2004 b). The UAM/RTM has also formed the basis of the further model development at the Freie Universität Berlin,
leading to the RCG-model, REM (Regional European Model)-Calgrid (Stern et al, 2003). LOTOS and REM-Calgrid were
sister models with intensive exchange of knowledge during their development.

The Eulerian air quality model EUROS (EURopean Operational Smog model) was originally developed at RIVM for the
modelling of winter smog ($SO_2$) episodes in Europe (van Egmond and Kesseboom, 1981). Later on, the model was used for
simulating various air polluting compounds in the lower troposphere over Europe, such as ozone  and Persistent Organic
Pollutants (POP's) ( Jacobs and van Pul (1996). The development of EUROS is described in  De Leeuw and van Rheineck
Leyssius (1990), van Rheineck Leyssius et al. (1990), Van Loon (1994, 1995), Hammingh et al. (2001) and Matthijsen et al.
(2001).



In both LOTOS and EUROS, aerosols were included around the year 2000 to simulate the inorganic secondary aerosols SO4, NH4 and NO3. (Schaap et al., 2004a; Erisman and Schaap, 2004; Matthijsen et al., 2002) and carbonaceous aerosols (Schaap et al., 2004b). In addition, data assimilation was implemented in LOTOS (Van Loon et al 2000) and EUROS (Hanea et al. 2004) in collaboration with the same research group at TU Delft. Since the two models had a similar structure and comparable

application areas, based on strategic and practical reasoning, RIVM and TNO agreed to collaborate on the development of a single chemistry transport model: LOTOS-EUROS. During 2004 the two models were unified which resulted in a LOTOS-EUROS version 1.0 (Schaap et al., 2005, Schaap et al 2008).

## 2.2    From LOTOS-EUROS v1.0 to LOTOS-EUROS v2.0

After release of version 1.0 in 2005 the LOTOS-EUROS model has been developed further to be able to (better) respond to new science and policy questions. We have retained the model's specific feature using a dynamic boundary layer approach in the vertical, handling vertical mixing in a different way than other models and enabling to apply the model over long time spans. The development was facilitated by the increasing quality and detail of input data, better process knowledge, increased computational capacity, and advance of remote sensing data. Societal challenges and political attention to adverse health

impacts, biodiversity loss and climate change have set the scene for new developments and applications.

The new EU legislation for particulate matter initiated strong interest in establishing the levels and origin of  particulate matter during average and episodic conditions.  The new European monitoring capacity for PM10,  and  later on also PM2.5, clearly revealed a systematic gap between observed and modelled concentrations. To improve the model skill for particulate matter

the parametrizations for the formation, emission and removal of individual components were revisited. To improve the modelling of secondary inorganic aerosol updates were made in the schemes for heterogeneous chemistry, cloud chemistry based on Banzhaf et al. (2012 and dry deposition (Zhang et al., 2001). To estimate the natural contribution to PM levels the source parametrization for sea salt was updated (Manders et al., 2010) and the impact of wild fire emissions explored (Martins et al., 2012).  This process also led to the introduction of mineral dust in LOTOS-EUROS with emission parametrizations for

road resuspension, agricultural land management and wind erosion or desert dust (Schaap et al., 2009; ). Specific source contributions from e.g. brake wear were addressed with new inventories for tracer components such as copper (Denier van der Gon et al., 2007). Although elemental carbon levels were modelled satisfactorily (Schaap and Denier van der Gon, 2004b; Hendriks et al., 2013), a major challenge remained for organic material. Although several schemes for the formation of secondary organic material were tested, no satisfactory model  parameterization is available yet. The development of the

Volatility Base Set (VBS) approach (Donahue et al 2006, 2009) seems the most promising approach has been implemented in LOTOS-EUROS, but its results still depend heavily on assumptions made. Towards understanding the origin of PM a labeling tool was implemented in LOTOS-EUROS (Kranenburg et al 2013), which enables to quantify the contributions of user specified emission sectors and regions to the modelled mass. Complementary to the model development, emission inventories





(e.g. Kuenen et al 2014) have improved in aspects such as resolution, spatial allocation, consistency and completeness. These inventories have been tested consistently with LOTOS-EUROS (e.g. Timmermans et al., 2013). In short, major advances have been made to model particulate matter, although the systematic bias has not been solved yet, mostly due to the challenges remaining for organic aerosol.

Current reactive nitrogen emissions to the atmosphere are estimated to be up to four times higher than pre-industrial levels and result in a cascade of environmental effects, including adverse health impacts through ozone and particulate matter formation and a loss of biodiversity through eutrophication and acidification of soils and surface waters (Fowler et al., 2013). Prior to the unification of LOTOS and EUROS most attention was given to the formation of secondary inorganic aerosol (e.g. Erisman

and Schaap., 2004; Schaap et al., 2004a). Over the following years focus shifted to analyzing (the origin of) episodic PM levels with high ammonium nitrate levels (Hendriks et al., 2016b). Reducing ammonia emissions can be effective, as long as it is not present in a large excess (Banzhaf et al., 2013; Hendriks et al., 2016b). Furthermore, the model system has also been intensively used to assess land use specific nitrogen deposition and subsequent critical load exceedances for Germany (Builtjes et al., 2011; Schaap et al., 2017). Continuous development has been performed on the deposition modelling including e.g. the

compensation point for ammonia (Wichink Kruit et al., 2010, 2012a) and droplet saturation effects for wet deposition (Banzhaf et al., 2012). These developments have resulted in a much larger consistency of the modelled air concentrations and wet deposition fluxes with observations. Still large uncertainties exist in the atmospheric budget of reactive nitrogen species, especially also in relation to ammonia (Sutton et al., 2013). This is explained by the short atmospheric lifetime and thus high spatial and temporal variability in ammonia levels combined with a lack of high quality monitoring capacity and large

uncertainties in emission distributions. Detailing the temporal emission variability based on meteorology and agricultural practices is pursued to improve the model skill to reflect the intra annual variability of ammonia (Hendriks et al., 2016; Kranenburg et al., 2017). Currently, satellite products for ammonia are emerging, which show a great promise for the validation of the LOTOS-EUROS model and its emission information (van Damme et al., 2014). Besides nitrogen deposition, specific attention has been given to the exposure of terrestrial ecosystems to Ozone (i.e. Phytotoxic Ozone Dose - PODy) (Bender et

al 2017) and heavy metals (Nickel and Schröder, 2016). Figure 1 illustrates  deposition and PODy applications  of LOTOS-EUROS over Germany.

The modest computational demand for running LOTOS-EUROS enables to perform many or long term scenarios. With respect to climate change air quality interactions the LOTOS-EUROS model has been used to evaluate scenarios assuming climate

change, energy policies and air quality mitigation as well as land use change. At first, these were addressed separately. The LOTOS-EUROS model was connected to the regional climate model RACMO-2 (Meijgaard et al 2008) and transient scenario simulations (1989-2100) downscaling global climate scenarios were performed to assess the feedback of climate change on air pollutant concentrations. These simulations showed a significant climate penalty on ozone levels (Manders et al, 2012), whereas none was quantified for particulate matter (Mues et al., 2013). Also a semi-online coupling between RACMO2 and



LOTOS-EUROS was established (Baklanov et al 2014) and contributed simulations to AQMEII-2 (Brunner et al 2014, Im et al 2015a,b).The potential impact of future wide spread biomass plantations on ozone distributions was highlighted by Beltman et al (2013). A recent scenario study for ozone showed that the impact of climate policies largely dominates over the concurring impact of land use change and that climate change might counterbalance the impacts of energy policies for ozone (Hendriks et al., 2016a). So far, common understanding is that except for ozone the impact of emission reduction largely exceeds the impact of climate change. However, dynamic evaluation of LOTOS-EUROS shows that it does not fully capture the impact of hotter and dryer summers as occurred in 2003 in Europe on $PM_{10}$ levels (Mues et al., 2012) indicating that this issue is not fully resolved. Dedicated energy transition scenarios targeted renewable energy generation with the EnerGEO project. A study to the impact of shifting temporal profiles of energy production facilities under a scenario with renewable energy deployment showed that the issue of increasing variability is relevant for air quality during the transition phase (Hendriks et al., 2015). In addition, the impact of a transition to a hydrogen economy was evaluated (Popa et al.,2015). To evaluate the ability of LOTOS-EUROS to perform scenario studies the model was used to evaluate air quality trends over the past 25 years (Banzhaf et al., 2015, Collette et al 2016).

An important development for LOTOS-EUROS was the participation in the EU-FP6 project MACC and its successors. This project meant to prepare for the operational Copernicus Atmosphere Service (CAMS) which is a European contribution to GEOSS. Access to ECMWF analyses and forecasts allowed to use LOTOS-EUROS for the provision of a daily air quality forecast over Europe and the Netherlands, thereby outperforming and replacing the Dutch statistical models (Manders et al 2009). The air quality forecast was shown to have considerable skill for the first 96 hours (de Ruijter de Wildt et al., 2011). In addition, time resolved information for boundary conditions and e.g. fire emissions became available. Currently LOTOS-EUROS is part of the regional air quality forecasting and analyses service within the Copernicus Atmosphere Monitoring Service (CAMS). This service provides operational forecasts and analyses of a.o. ozone, nitrogen dioxide and particulate matter based on an ensemble of seven models (Marécal et al, 2015). The LOTOS-EUROS forecasting service is operated by the Royal Dutch Meteorological Institute (KNMI) warranting an availability of at least 98%. The forecasting system (Figure 2) provides 96-hr forecasts of air quality twice per day. The national service is now delivered through nesting within the European scale CAMS service. In addition to air pollutants, birch pollen concentrations are forecasted (Sofiev et al 2015). Currently, near real time surface observations of ozone, $NO_2$ and PM and satellite based OMI tropospheric $NO_2$ column data are assimilated to provide near real time-analyses of air quality (Denby et al., 2008, Curier et al., 2012). Assimilation strategies for other components such as $SO_2$ (Barbu et al., 2009) and AOD (Segers et al.,2010) have also been investigated, but are not yet fully operational. The access to global input data has allowed to extend the area of operation to other regions in the world (e.g. Timmermans et al 2017). Currently, operational forecasts are delivered for China (http://www.marcopolo-panda.eu/forecast/) and northern Africa (http://sds-was.aemet.es/forecast-products/dust-forecasts/compared-dust-forecasts).



## 3    Model development strategy

### 3.1    Philosophy

The basic philosophy is that the model is both state of the art and reliable, since it has to be used for operational air quality forecasts and policy support applications. Scientific developments are included in the base version of the model after thorough

testing of their benefits. This is the reason that we were and still are reluctant in the use of for example the VBS approach, for which the outcome depends heavily on assumptions. Furthermore, the level of detailing of a process should match the general level of accuracy of the model, given uncertainties in e.g. meteorological and emission input, model resolution and the accuracy of other modelled processes. On the other hand, the use of the compensation point for NH3, the source apportionment (labelling) approach and data assimilation are features which distinguish LOTOS-EUROS from other models.

### 3.2    Version control

To be able to perform operational calculations, respond to customer requests and to be able to explain differences in model behaviour  a development system has been adopted at TNO. The idea is that the impact of every model development, even as small as an alternative calculation of a meteorological parameter, is traceable. Hence, to document the impact of a development

a benchmark test has to be performed to document the isolated impact of an alteration in the code. Although this approach add additional workload, it is crucial for quality control, scientific understanding and documentation.

The model development is performed in projects based on a single base version. For each new development the developer adapts particular pieces of code, which are saved in a separate folder dedicated to the development project. The base code is

combined with the altered code in the project folder to build an executable of the model. In this way several developments can be performed at the same time. Compilation of the model code takes place as a part of the initialisation of a (test) simulation. The model code, executable and simulation settings file are copied to the model output directory so that every simulation can be reproduced exactly.

Annually, the developments and their impacts are reviewed to select the functionalities which need to be maintained in a new base version. After completion of the new model version the full benchmark test is performed to perform quality assessment and quality control (QAQC) and assess the model performance in a statistical way. Previously, the benchmark test covered the year 2006. With the completion of the open source version of LOTOS-EUROS also the benchmark test is updated covering 2012. The new benchmark test described below was chosen as the measurement data availability has  increased over recent

years in Europe. Moreover, new input data become available (e.g. CAMS boundary conditions) for recent years, but are normally not provided for historical years. Hence, it appeared practical to start performing the benchmark tests for a more recent year.



### 3.3    Model evaluation

A major aspect of air quality modelling has always been model performance/ model validation (e.g. Fox, 1981, Rao and Visalli, 1981). Each new model version as well as (dedicated) codes for use in particular projects are evaluated in comparison to observations using standard statistical parameters. As such the operational evaluation as defined by Dennis et al. (2010) is executed very often. Dynamic and diagnostic evaluations are much more effort to carry out and are performed occasionally.

For example, Mues et al (2012) addressed the ability of the model system to reproduce the summer of 2003, whereas Stern et al., (2008) showed general difficulties of capturing pollutant distributions during very stagnant conditions. Recently, Banzhaf et al. (2015) showed that the model system is able to reproduce non-linear behaviour observed in trends of secondary inorganic aerosol across Europe. In addition to the traditional model evaluation strategies a new perspective on assessing model performances is through data assimilation. Data assimilation techniques can be used to detect shortcomings in model

descriptions and input data (see below).

Apart from our own validations, LOTOS-EUROS participates as much as possible in model comparison studies in which the model performance is assessed in comparison to its peers. These exercises have increased the interaction with colleagues through dedicated discussions and exchange of experiences and have contributed to the detection of model flaws and

subsequent improvement, in particular during the first studies. The first intercomparison was launched within EUROTRAC-GLOREAM (Hass et al., 1997), which was extended into the review of the EMEP model in 2004 (van Loon et al., 2004). These studies were the basis of the CITYDELTA and EURODELTA studies in which the robustness of model responses to emission changes was studied with an ensemble of 7 chemistry transport models (van Loon et al., 2007). In addition, LOTOS-EUROS took part in intercomparison studies from COST (Stern et al., 2006) and AQMEII phase 1 (Solazzo et al 2012a,

2012b), phase II (Im et al 2015a,b) and the ongoing phase III   and has recently taken part in the EURODELTA phase III (Bessagnet et al 2016, Garcia Vivanco et al 2017)  and EURODELTA trend analysis in which several models have simulated the period 1990-2010 (Colette et al 2017). Through the intercomparison studies the team also benefits from (new) analysis techniques and expertise from a range of scientists. Such an  innovation in model evaluation is applied in Solazzo and Galmarini (2016)  who analysed results in a new investigated behaviour of models on different time scales (seasonal-synoptic-daily-

hourly). The model intercomparison studies have demonstrated that a model ensemble generally provides the best performance in comparison to observations, (e.g. Vautard et al 2009) as compared to  although this requires that models or model versions are independent (Potempski et al 2009).



## 4    Model description

This section briefly describes the most important features of the model version v2.0. A more elaborate description can be found in the model documentation (Manders et al 2016a)

### 4.1    Domain, grid

LOTOS-EUROS is a regional model on a regular longitude-latitude grid. It is typically used with a resolution of 0.5x0.25 ° on a domain covering most of Europe and the Mediterranean sea, but can be applied anywhere and with arbitrary grid resolution, provided that the horizontal resolution is not smaller than about 3 km. This is related to the vertical structure which is quite special to the model. In the vertical, the model consists of a static surface layer of 25 m, a dynamic layer representing the mixing layer, and three dynamic reservoir layers covering the lower 5 km of the troposphere. In earlier version two reservoir

layers extending up to 3.5 km were often used, but also extension to 10 km has been done. The mixing layer is defined by the mixing height of the meteorological input and is interpolated in time. It has as a minimum height of 50 m. The lower two reservoir layers are equally thick with a minimum of 500 m, and the third reservoir layer is designed to be 1500 m thick in order to extend from 3.5 to 5 km, unless the mixing layer is very thick. In the mountains (or the tropics), the mixing layer may extend to more than 3500 m and the top of the model is extended to fulfil the requirements for minimum thickness of the

reservoir layers. The large advantage of the current vertical structure is that it makes the model very efficient  in terms of solving the chemistry, the most time-consuming process. However, if higher resolution is desired, the horizontal and vertical dimension could be out of balance  for the used parameterizations and  more layers have to be added.

### 4.2    Tracers/species

The model was primarily aimed at air pollution. It models the gas-phase chemistry of ozone ($O_3$, $NO_x$, VOC, isoprene, CO), and gas-phase/aerosol conversions of  sulfur components ($SO_2$, $SO_4$), reduced nitrogen ($NH_3$, $NH_4$) and  oxidised nitrogen ($NO_3$). It also explicitly models other primary PM constituents (elemental carbon, organic carbon, other primary PM, mineral dust, sea spray, heavy metals like Cr, base cations like Ca and Mg).  There is the possibility to calculate secondary organic aerosol with a 1-dimensional VBS scheme. For climate applications, $CO_2$ can be modelled as a tracer. The required groups of

tracers for a simulation can be easily selected.

### 4.3    Chemistry

The gas-phase chemistry is a condensed version of CBM-IV (Gery et al 1999), with some modifications in reaction rates and can be found in (Manders et al 2016). A kinetic pre-processor is used which makes it relatively straightforward to add or

modify chemical reactions. For the secondary inorganic chemistry  Isorropia II (Fountoukis and Nenes 2007) is used and



heterogeneous chemistry on wet aerosol (Wichink Kruit et al 2012b). Also a pH-dependent cloud chemistry is used (Banzhaf et al 2012). We have included the option to use the one-dimensional VBS approach (Donahue et al 2006) with nine volatility classes in a very conservative way. Anthropogenic emissions of primary organic material are assigned to the four lowest volatility classes and an additional 1.5 times this mass is assigned to the higher five classes. Isoprene and VOC contribute to

SOA formation but the impact of terpene is currently not taken into account. Although the impact of the latter is significant due to the relatively high mass of terpene as compared to isoprene, emissions and conversion rates are rather uncertain (Bergström et al 2012, Zhang et al 2013), therefore the VBS is not used by default.

## 4.4    Meteorology

LOTOS-EUROS has interfaces to several meteorological model output sets. Apart from temperature, wind fields, boundary layer height, cloud cover and vertical distribution, incoming radiation and rain/snow, specific humidity also surface properties (soil moisture), sea surface temperature and snow/ice coverage are required. These are relevant for sea spray emissions, dust emissions and deposition velocities. By default, it uses 3-hourly ECMWF short-term forecasts meteorology, interpolated to hourly values, but the model has also been run with WRF and HARMONIE meteorological input, and has been coupled semi-

on line to the regional climate model RACMO2 (Meijgaard et al 2008). When not all meteorological fields are available, e.g. soil water content, representative average values can be used. Friction velocity and Monin-Obhukov length are calculated on-line based on the land use parameters (roughness length) of LOTOS-EUROS and  wind speed, solar zenith angle and cloud cover.

**4.5**    **Emissions**

Emissions of biogenic NMVOC, mineral dust (wind-blown dust and resuspension caused by traffic and agricultural practices) and sea salt are calculated on-line using meteorology-dependent relations described in Schaap et al. (2009). Sea salt emissions are calculated according to Mårtensson et al. (2003) and Monahan et al. (1986) based on 10m wind speed and sea surface temperature. Hourly emissions from forest fires are taken from the MACC global fire assimilation system (Kaiser et al., 2012).

Emissions of NO from soils was included using the parameterization depending on soil type and soil temperature from Novak and Pierce (1993). For the emissions of isoprene and terpene the MEGAN routine is available (Guenther et al 2006), but for Europe a slightly different approach is taken using a tree species database, as described in Beltman et al (2013). LOTOS-EUROS calculates on-line dust emissions based on the sand blasting approach by Marticorena &Bergametti (1995) and including soil moisture inhibition, potential sources and a soil map (e.g. Mokhtari et al 2012). Since soil emissions are very

sensitive to region-specific local conditions the optimal settings depend on the region.



The TNO-MACC-III emission database (available for 2000-2011) was used for the anthropogenic emissions of $NO_x$, $SO_2$, $CH_4$, CO, NMVOC, $NH_3$, $PPM_{2.5}$ and $PPM_{10}$. This database is an update from the TNO-MACC-II dataset (Kuenen et al., 2014) and contains high-resolution (0.125 x 0.0625 ° lon-lat) emission information based mainly on official country reporting of national emissions to UNECE and the EU. Emissions are presented in aggregated source categories (SNAP levels) as a total

annual sum for each country. These have been disaggregated spatially using actual point source locations and strengths as well as several proxy maps for area sources (Kuenen et al., 2014). The temporal disaggregation of emissions is done using sector-specific monthly, daily and hourly time factors and include temperature-dependent factors for CO and VOC to account for a cold start for passenger cars. In the vertical fixed emission profile per SNAP code are used (following the approach of EURODELTA, Thunis et al., 2008, see Manders et al., 2016a for details). Scenario factors on specific countries or source

sectors can be defined in a separate file and dedicated emission sets can be ingested without changes to the code.

### 4.6    Land use

Land use data are an important input parameter to model biogenic emissions of NMVOC, emissions of mineral dust and NO from soils. Moreover, the land use type determines dry deposition characteristics of atmospheric species. We use the

Corine2000 Land Cover database (EEA, 2000) with a grid resolution of 0.0167° (~1.9 x 1.2 $km^2$ at 50° North) in longitude and latitude over Europe. This database is complemented with the distribution of 115 tree species over Europe (Koeble and Seufert, 2001). The combined database (which can be updated with Corine2006, EEA 2007) has a resolution of 0.0166° x 0.0166° which is aggregated to the required resolution during the start-up of a model simulation. Each grid cell in LOTOS-EUROS is characterised by the fraction of several types of land use in that particular grid cell. A land-sea mask at 1/112 degree

lon-lat        resolution         (based         on         the         World         Waterbodies         GIS         map, http://library.duke.edu/data/files/esri/esridm/2013/world/data/hydropolys.html) is used to distinguish land area, inland water and seas in more detail. The Black Sea, Caspian Sea and Sea of Azov, labelled "perennial inland water" in the World Waterbodies database where put to "ocean or sea" instead, since they are so large that waves may develop that have significant impact on deposition velocity.

### 4.7    Deposition

Wet deposition is divided over in-cloud and below scavenging. An in-cloud scavenging module based on the approach described in Seinfeld and Pandis (2006) and Banzhaf et al (2012) is included, the previous simple below-cloud scavenging approach with scavenging coefficients for aerosols and gases (Simpson et al. 2003) was left for backward compatibility. For

dry deposition, a resistance approach is taken. The parameterizations by Zhang et al (2001) are implemented for particles, for gases the DEPAC module is used (Van Zanten et al 2010). Dry deposition velocities are not only used for the calculation of removal of species, but also to translate concentrations in the surface layer to concentrations at observations height (2.5 m) by



using a constant flux approach in the lowest layer. For NH$_3$ a compensation point approach is implemented (Wichink Kruit et al., 2010, 2012a).

## 4.8    Boundary conditions

Boundary conditions are an  essential part of regional models, in particular for components with long lifetime like CH4 and high hemispheric background concentrations like O3. As a basis set, the climatologies by Isaakson and Logan are chosen, with the Mace Head correction for ozone as provided by EMEP (based on Derwent et al 2007). More detailed boundary conditions can be provided by global models (e.g. TM5) or the global systems in the CAMS system. For operational applications, CAMS

boundaries are used for several components. Near real time boundary conditions originated  from the global MOZART model in the past and more recently the CIFS system (Flemming et al 2015). When model versions are updated or new data become available for assimilation the signature of the boundary conditions may change significantly, with large impact on e.g. ozone background levels and thus on model performance of LOTOS-EUROS. Next to near real-time boundary conditions also reanalysis data are available from CAMS that provide longer and consistent series. Which set of  boundary conditions is used

depends on the application.  For high-resolution applications we use a LOTOS-EUROS simulation on a larger domain to nest our smaller high-resolution domain.

## 5    Benchmark for 2012

## 5.1    Set-up

To evaluate LOTOS-EUROS model performance, we use a simulation for 2012 with input datasets specified in Table 1. These inputs are commonly used in LOTOS-EUROS studies. In operational applications, often boundary conditions from global models are used (e.g. MACC products) but their quality depends on the year they are produced. Since their impact is rather large, for this validation study we have chosen to use climatological boundary conditions and the Mace Head correction for

ozone based on Derwent et al 2007,  provided by EMEP.



## 5.2    Observation data and evaluation

The principal source of observation data used in this benchmark is the EMEP network (Tørseth et al., 2012), which provides data from rural and remote measurement stations on an hourly or daily basis. Time series for concentrations of $O_3$, $NO_2$, $NH_3$, $SO_2$, $PM_{10}$, $PM_{2.5}$, EC, Na, dust, $NO_3$, $TNO_3$, $SO_4$, $NH_4$ and $TNH_4$ are available and used for the model evaluation. For aerosol

composition data the time series for the EMEP aerosol samplers as well as $PM_{10}$ samplers are combined. In addition, chemical analysis of rain water used to evaluate the modelled wet deposition fluxes. Stations located at an altitude above 700 m are not considered, and data flags are taken into account by excluding all data with irregularity. Moreover, a visual screening of the data was performed to assess the quality of the data. Obvious reporting errors mostly concerning unit conversions were found and corrected when confirmed through checks with earlier data downloads. A data availability of >75% was chosen for a

station to be included, which is rather strict but prevents comparison of stations that have operated for only part of the year, omitting for example a full season. Only for Na and EC the availability criterion was set to 50% since these require laboratory filter analysis which is for most stations done less often. The observation data set is frozen as the dataset needs to be used for the validation of future versions for a number of years ahead, and EMEP may update its data. The operational model evaluation is carried out through the calculation of standard statistical measures such as RMSE, Bias and spatial and temporal variability.

Average correlation and bias are presented to reduce the size of the tables, for performance on individual stations we refer to the validation report (Manders et al 2016b).

## 5.3    Results

Annual average modelled concentrations of $O_3$, $NO_2$, $NH_3$, $SO_2$, $PM_{10}$, $PM_{2.5}$, EC, sea salt, $NO_3$, $SO_4$ and $NH_4$ are shown in Figure 3. They represent the common features related to emission hotspots for emissions related to combustion (large cities,

densely populated areas for $NO_x$, EC) and agriculture (NH3, most prominently North Western Europe, southern Germany,  Po Valley and Bretagne). For $SO_2$ and $SO_4$ Poland and South-Eastern Europe are dominant, where coal use is relatively large and desulfurization is not applied everywhere. OC concentrations are larger in areas with wood combustion. Ozone shows a pattern that is related to both temperature (increasing concentrations for southern latitudes) combined with lower concentrations in areas with high NO for ozone titration. The secondary inorganic aerosols show a pattern that is a smoothed version of the

precursors due their longer lifetime and resulting transport distances. Sea salt clearly shows a strong gradient near the coast, with generation over sea and rapid removal by deposition over land. The zero boundary conditions for sea salt result in unrealistic values at the western boundary of the domain. Not using a dust boundary condition leads to too low $PM_{10}$ concentrations in the southern boundary of the domain.

Figure 4 illustrates the time correlation for ozone and $PM_{10}$ for a station in the Netherlands. Vredepeel is a rural station in a region with intense agriculture and part of the time influenced by the Ruhr area. Ozone is slightly overestimated in summer





but in general in close agreement with observations, with periods of elevated concentrations during warm conditions. For $PM_{10}$ LOTOS-EUROS generally underestimates the concentrations with a few μg/m$^3$ but clearly underestimates concentrations in a few peak episodes. These are mostly related to cold and stagnant winter episodes with more emissions and less ventilation, cold and stagnant conditions are generally an issue for air quality models as input for wind speed, stability and boundary layer height from numerical weather models is not always accurate (Bessagnet et al., 2016). In addition, cold weather leads to more emissions for residential heating, which is not taken into account with the emission time profiles used here.

Table 2 reveals that the spatial correlation is very good for all components, with 0.68 for ozone as the lowest values and values up to 0.95 for $NH_3$. Spatial correlations for annual mean ozone are relatively poor due the following aspects, which differ per region and station. The annual cycle is strong in the South and weak in the North of Europe, very low night time concentrations for some stations are not captured by the model, there is an overestimation of baseline concentrations at the western part of the Iberian peninsula due to high boundary conditions and a relatively poor reproduction of the annual cycle for Scandinavian and Baltic stations. For summer daily maximum and 8-houly maximum, for which these effects are much reduced, the spatial correlation is indeed very high, This also influences the average performance in time correlations, which are quite low, although for many individual stations high correlations for daily maximum and summer 8-hourly maximum are found (e.g. 0.75 for Kollumerwaard). In contrast, despite the excellent spatial correlation, the time correlation for $NH_3$ is one of the poorest. The reason is that emissions of $NH_3$ depend strongly on meteorology in reality (favourable circumstances for manure application, temperature-dependent stable emissions) and is deposited quickly. In the simulation long-time average time profiles were used for emissions, thus day-to-day variations in emissions were not taken into account. In section 6.3 this is explained further. Due to different uncertainties per model component (emissions, chemical conversions, chemical interactions between species, deposition) all modelled species have a different behaviour.

Figure 5 shows modelled annual total wet and dry deposition of oxidized and reduced nitrogen and oxidized sulphur. For dry deposition fluxes, patterns broadly reflect the emission patterns, smeared out by transport. For wet deposition areas with large precipitation sums (coast and mountain areas) are additional hot spots, particularly well visible for $NO_y$. The comparison with monthly mean rain water concentration observations from the EMEP network is in Table 2 and Figure 3. The spatial correlation of concentrations in rain water is very good with values around 0.8, but the temporal variability is poorly reproduced. Figure 4 illustrates this for Kollumerwaard, a rural station in the North of the Netherlands, close to the Wadden Sea. Annual average values are underestimated for most station, although in the time series per station in some months overestimations and in other monts underestimations occur, resulting in modest time correlation. For wet deposition samples only 12 values per year are available. This limited set of data points drastically reduces the significance of the correlation and eventual outliers are not easily identified. Wet and dry deposition process descriptions are relatively poorly constrained by direct measurements of deposition velocities and scavenging rates. Additional inaccuracies arise from the sometimes local character of rain which is



not captured by the scale of the model and meteorological model input. A detailed discussion and model intercomparison of wet and dry deposition can be found in Garcia Vivanco et al 2017.

## 6    Research areas and innovative applications

The core of the model is a reliable and efficient calculation of gas-phase and aerosol components. Here we highlight the model's special functionalities and applications. The source apportionment and particle number modelling features are not part of the open source version of LOTOS-EUROS v2.0 as they are in the research phase and the code is updated relatively often. The data assimilation system is separate shell around the model and not part of LOTOS-EUROS itself.

### 6.1    Data assimilation system

A range of techniques can be used to assimilate, or combine, observations with modelled concentration maps for analyses of air pollution situations. Passive data assimilation methods include statistical assimilation techniques such as optimal interpolation methods, residual kriging methods, regression and multiple regression techniques, e.g. Blond et al. (2003); Horálek et al. (2005; 2007). These assimilation techniques are most often applied 'off-line' in a sense that the model output is

combined with observations as a post-processing step. Modelled air pollutant distributions of LOTOS-EUROS and its predecessors have regularly been used to investigate new offline methodologies (e.g. Kassteele and Velders, 2006; Kassteele et al., 2006; Hamm et al., 2015). However, these techniques do not provide information on uncertain model parameters, such as emissions, and are often difficult to apply within an operational forecasting system. For air quality forecasting applications the positive impact of the data assimilation of observations is usually quickly lost (within 1 day) when only updating the initial

state.

Additional updates of emissions through active data assimilation have been shown to lead to improvements that last longer (Lahoz et al., 2007; Timmermans et al., 2009; Curier et al., 2012). To allow parameter estimation and further improvements of forecasts, emission monitoring assimilation strategies for air pollutants were developed since the late nineties. Central to

the assimilation of observations with LOTOS-EUROS has been the development of an Ensemble Kalman filter system (EnKF) (Evensen 1994), which allows updates of model parameters through the assimilation of observations.  LOTOS-EUROS with EnKF has been applied in a number of applications directed at ozone, sulphur dioxide and/or nitrogen dioxide (e.g. van Loon et al., 2000; Hanea et al., 2004; Barbu et al., 2009; Van Velzen et al 2010, Curier et al., 2012) as well as particulate matter (Denby et al., 2008 and Figure 6) and volcanic ash (Fu et al., 2015). Besides in-situ data, satellite tropospheric NO2 column

observations (OMI: Curier et al., 2016) as well as Aerosol Optical Depth (AOD)  (SEVIRI: Segers et al., 2010)  have been



successfully assimilated in the LOTOS-EUROS model. Figure 6 illustrates the improvement in PM10 forecasting by assimilating $PM_{10}$ ground observations.

The assimilation system has also been used to assess the added value of future satellite instruments through so-called Observation System Simulation Experiments (OSSE's, Timmermans et al., 2015). The added value of a future observation system is investigated by producing synthetic observations using a different model simulation (nature run), and assimilate these data in the model. Experience has been obtained for potential new instruments for aerosol optical depth (Timmermans et al., 2009) and nitrogen dioxide. To improve the parameter estimation with respect to emission strengths a new direction is

to explore variational assimilation techniques that does not require the implementation of the adjoint model of LOTOS-EUROS (Lu et al, 2016). Also remote sensing has evolved from only detecting AOD to retrievals of microphysical properties like aerosol size and number, which may be assimilated in the future.

## 6.2 Source apportionment

LOTOS-EUROS includes a source apportionment technique to track the origin of air pollutants (Kranenburg et al., 2013). This module tracks the contribution of sources through the model system using a labeling approach similar to Wagstrom et al. (2008). The emissions can be categorized and labeled in several types of categories (e.g. countries, sectors, time of emission) before the model is executed. The labeling routine is implemented for both inert and chemically active tracers containing a C, N (reduced and oxidized) or S atom. Among other applications, this module was used to study the origin of particulate matter

in the Netherlands (Hendriks et al., 2013), changing source receptor relations for energy scenarios in Europe (Hendriks et al., 2015) and particulate matter sources in Chinese cities and regions (Timmermans et al 2016). Another application was to investigate the sensitivity of the OMI instrument to $NO_x$ emission sources in Europe (Schaap et al., 2013, Curier et al., 2014). The module was also recently used on a high resolution application for the Netherlands to determine the influence of several source sectors (e.g. shipping, road transport and residential heating) to PM concentrations at city scale. Figure 7 illustrates the

case for Rotterdam, giving detailed insight in the contribution of several sectors at specific air concentration levels. By splitting the information in contributions of source regions and sectors, the potential of local measures to reduce air pollution can be quantified and could be used for local measures when poor air quality is forecasted.

## 6.3 Emissions modelling

Annual totals of emissions are known relatively accurately due to emission reporting obligations and the spatial distribution can be derived from e.g. population density and road networks. Assessments of emission information can be done by comparing




model outcomes with satellite or ground-based observations. This method can be used to check the total quantity of emissions (Curier et al., 2014) or mislocations of sources.

Relatively much improvement in model performance comes from improving the timing of emissions. The distributions of emissions over time are poorly represented by the default time profiles that are used, as they are based on average annual cycles. In a study focused on Germany, timing of emissions was improved by using traffic counts for the road transport sector, electricity demand for the power plant emissions, and using air temperature for redistributing residential combustion emissions (concept of heating degree days), leading to better model performance (Mues et al., 2014). Also ammonia emissions from agriculture are strongly depending on meteorology (temperature, temperature sums, rain). Figure 8 shows an example where local legislation and meteorology were taken into account in the NH3 emission time profiles (Hendriks et al 2016b), extending the work of Skjøth et al. (2004;2011). Such advanced timing of anthropogenic emissions is accounted for in a pre-processor of the emissions and not part of the LOTOS-EUROS code. The relatively recent availability of new meteorological variables for soil conditions (moisture, temperature, evaporation) can further improve the timing and amount of natural emissions from soils (NOx, Dammers 2013, not part of LOTOS-EUROS v2.0), and mineral dust, see below) which are calculated on-line.

## 6.4 Aerosol modelling improvement

There is still a gap between observed and modelled PM10 concentrations. For some species the correspondence with observations is quite good. For others it is more uncertain. Recent developments in aerosol modelling in LOTOS-EUROS include the implementation of the VBS scheme, an update of the modelling of desert dust and the implementation of a module including nucleation, condensation and coagulation to describe particle number concentration and evolution.

LOTOS-EUROS v2.0 has now an implementation of the VBS scheme for **secondary organic aerosols** as a promising method. The use of previous existing parameterizations (e.g. SORGAM, Schell et al. 2001) did not lead to a significant improvement of model performance in LOTOS-EUROS and SOA chemistry was therefore not applied. Model intercomparison studies with models that did include VBS or SORGAM justified this decision. Depending on settings of the model, still a wide range of results can be produced (Bergström et al 2012). In its current conservative implementation in LOTOS-EUROS, differences with and without VBS are in the order of less than 1 $\mu g/m^3$ (less than 3%) on the annual average concentrations and therefore VBS is not used by default. But when settings and reaction rates become more well established the VBS scheme can be activated or extended.

**Mineral dust** is the dominant contributor to $PM_{10}$ in some areas of the world and during specific events. It is generated from deserts but also from bare agricultural soils outside the growing season. Good modelling of mineral dust emissions is a challenge and results from different models are easily a factor ten apart. This is because generation of windblown dust is very



sensitive to local wind speed and to regional and local roughness length (Menut et al., 2013b) and soil characteristics (Mokhtari et al., 2012). Therefore all models use tuning factors to come to optimal settings for the region of interest. LOTOS-EUROS has been used to model dust from the Gobi desert (Timmermans et al 2016), from the Sahara (LOTOS-EUROS is now part of operational SDS-WAS dust warning system since November 2016, see also Figure 9) and for European dust events

(unpublished). LOTOS-EUROS is one of the few models that explicitly includes dust from road resuspension and agricultural activity (input in this moment only available for Europe).

LOTOS-EUROS can use the  M7 module (Vignati et al 2004) to model **particle number (PN) concentrations**, including the processes of nucleation and condensation of H2SO4 and coagulation of particles. Particle numbers are dominated by the size range of a few nm up to 300 nm which includes ultrafine particles (UFP, corresponding to PM0.1). UFP contribute little to total PM mass but they are relevant since they are abundant, can intrude deeply into the lungs with adverse health effects. Slightly larger particles in the range of 100-300 nm are relevant for climate modelling as they may grow towards the size of

cloud condensation nuclei  make them relevant for climate (e.g. Kulmala et al 2011, Paasonen et al 2013).In LOTOS-EUROS, M7's original nucleation scheme by Vehkamäki has been replaced by an activation type (Kulmala et al 2006) to be more representative of the boundary layer instead of the troposphere, leading to a better correspondence with observations. The model performance is best for areas that are dominated by anthropogenic sources.  For model validation, a description of emission input,  and application as background model for city-scale models we refer to  Kukkonen et al., 2016.  The modelled

size distribution does not match the observations very well, with an overestimation of small particles concentrations and an underestimation of large particles. This problem with size distribution is comparable to a similar model application with CAMx (Fountoukis et al., 2012). Nevertheless,  the overall  total modelled number concentrations are in the right range. It has also been applied at high resolution over the Benelux area (see Fig. 10), showing the large contribution of road and ship traffic to ultrafine particle concentrations. In general, UFP and PN modelling, as well as PN emission inventories, needed as input, are

a recent development where significant further research is needed.

## 7    Discussion and outlook

The decision to join the Dutch modelling capacity and unite LOTOS and EUROS has proven fruitful. After ten years of model development the  LOTOS-EUROS model is in good shape and the general performance of v2.0 is satisfactory. In model intercomparison studies LOTOS-EUROS falls well within the range of other models and is for some species among the best

performing models (see e.g. Bessagnet et al., 2016,  Im et al. 2015a, 2015b for the performance of recent previous versions). These studies also show that there is no single model that is best for all species. During its development from v1.0 to v2.0 LOTOS-EUROS has retained the original set-up with the efficient layer system and the model can be applied for both



operational forecasts and long-term climate and scenario applications. The labelling technique and data assimilation make LOTOS-EUROS stand out. The next step is to extend the community and the model version 2.0 was made available as an open source model. There are several lines of research for further improvement.

A large remaining issue is the general underestimation of PM mass by LOTOS-EUROS, a feature that is shared by most chemistry transport models. In v2.0 this underestimation of $PM_{10}$ was reduced as compared to previous versions, amongst others by a change in deposition velocity for arable land outside the growing season, new meteorological input data, and taking soil $NO_x$ emissions into account. To further improve upon this, we need to further develop several aspects of the model that we will discuss now. Model evaluation showed that secondary inorganic aerosols were underestimated on average, in particular

$SO_4$ and $NH_4$ and to a minor extent $NO_3$. Part of the $SO_4$ underestimation is related to PM peak episodes in winter, and thus related to issues with emission timing and poor representation of mixing during stagnant conditions. Also too inefficient heterogeneous chemistry could play a role in the underestimation by underestimating the conversion of $SO_2$ to $SO_4$. As indicated above, another part of the missing aerosol must come from secondary organic aerosol. The VBS approach seems a good starting point (Bergström et al 2012), but not all issues are solved yet. Next to the current 1-D VBS scheme, 2-D schemes

have been developed, not only taking into account the saturation vapour pressure but also the O:C ratio (oxygenation state) (Donahue et al 2011). In addition, the role of reactive nitrogen becomes more clear (Pye et al., 2010, 2015), which is not taken into account yet. These developments have contributed to a better description in other studies (Pye et al., 2013). But also using a better emission inventory for OC, including the condensable fraction of the aerosol directly in the emission inventory (Denier van der Gon et al., 2015 ) made a large difference in bridging the gap. The improvement of emissions is and will be one of the

focal points of  LOTOS-EUROS. It brings together expertise on emission inventorying and data assimilation. Improvements are possible as more detailed information  becomes available in terms of reporting, near-real time activity data, and satellite observation. Also dependency on meteorological conditions will be modelled more realistically in the future (e.g. soil NOx emissions, manure spreading, heating degree days). Natural emissions of $NO_x$ from soil are dependent on soil type, soil moisture, precipitation and evaporation, and temperature. An improved description of soil $NO_x$ emissions was developed

(Dammers,2013), making use of more detailed soil characteristics like soil moisture, but as yet this was not implemented in an official model release. More differentiated time profiles become even more important when going to higher resolution

More detailed land cover information is required for improved model performance, in particular now that the model is applied

over different regions in the world. A step forward would be the harmonization of all soil characteristics, vegetation maps and land use maps necessary for the calculation of  natural emissions and for deposition. A future development would be to include a roughness length map instead of a using fixed roughness length for land use categories (e.g. Menut et al., 2013b).  In addition, areas may be partly covered by vegetation during part of the year and bare during other seasons, which is not taken into account yet and leads to overestimations of dust emissions or deposition for some regions. Deposition on vegetation, pollen release



and release of biogenic VOC would benefit from vegetation indices that go beyond climatological growing season descriptions. A better incorporation of knowledge of local agricultural practices would be needed in order to determine when and where large areas of agricultural land are susceptible to wind erosion, and when dust emitting activities like ploughing or harvesting take place (Schaap et al 2009). Also satellite-based vegetation indices could be used. This would be a relevant development

in terms of modelling soil-biosphere-atmosphere exchange which is getting more attention in view of earth system modelling and understanding of chemical cycles of N or $CO_2$ budgets.

A recent development in regional-scale chemistry transport models is to run at high resolution (e.g. Collette et al 2014 at 2 km

resolution, Kuik et al 2016 at 1 km resolution) to describe strong gradients within cities. Next to representing strong gradients in cities, for some areas with intensive agriculture, it is desirable to calculate ammonia concentrations and deposition at this high resolution so that the model can be better used for regulatory purposes. LOTOS-EUROS is being developed to function at this high resolution. Because of the specific vertical structure of the model, with the mixing layer as first dynamical layer, this is not possible by simply increasing the horizontal grid resolution as this layer may easily become thicker than 1 km. Also

the vertical structure has to be adapted, with more vertical sublayers in the mixing layer. A version in which more vertical layers are implemented without losing the model's characteristic efficiency is under construction and will be closely related to the layers of the driving meteorological model. For high-reolution application meteorology at higher resolution than now available form ECWMF is required. Such high-resolution meteorology is available from regional models like WRF (Fast 2006, Grell 2004) and models operational at European weather institutes (e.g. HARMONIE, COSMO). Interfacing to these

meteorologies is under development. To overcome the intrinsic larger computation times at high resolution, the implementation of domain decomposition would be necessary to enable efficient parallel computing.
Alternatives for a high resolution of LOTOS-EUROS itself are the implementation of a plume in grid approach (Seigneur et al 1983, Karamchandani 2011, Rissman et al 2013) and the (off-line) coupling with plume or street models (Brandt et al, 2001, Kukkonen et al 2015). An intermediate solution for the calculation of annual average maps, has been demonstrated for the

Netherlands by combining the LOTOS-EUROS results with those of the OPS model (Van Jaarsveld, 2004; Sauter et al., 2015). Van der Swaluw et al. (2017) describe how model outputs were combined in such a way that contributions to concentrations in the Netherlands that stemmed from Dutch emission sources were obtained at high resolution from OPS and contributions from abroad were delivered by LOTOS-EUROS.

The development towards high resolution is largely a technical one. Societal needs will involve more than high resolution. These include questions related to more detailed health- and climate related scenarios, such as emission and transport of ultrafine particles and man-made particles including nanomaterials, emissions and monitoring of species related to shale gas production. Operational services like CAMS and new satellites that will be launched in the (near) future will push the generation and use of data streams to a next level. Not only air quality is relevant, but also derived quantities like forecasting





solar energy are emerging. Areas which receive much solar radiation in cloud-free conditions suffer most from dust. On the input side, near real time data on land cover, vegetation height etcetera could improve model performance. The increased level of detail that is represented in the denser network of ground-based and satellite observations can only be mimicked if the model input is in the same detail, since process parameterizations in the model are generic with as little tuning as possible.

The wealth of new input data, verification data, societal and scientific questions ensures that modelling of atmospheric composition is still a lively field of research and LOTOS-EUROS has the potential for further development to meet future needs.

**Code availability**

LOTOS-EUROS is written in FORTRAN90 and uses NetCDF libraries and python scripts. The open source version of LOTOS-EUROS can be obtained through http://www.lotos-euros.nl/open-source/index.php. Additional functionalities can be disclosed upon request (astrid.manders@tno.nl).

**Author contributions**

A. Manders: Preparation of text with contribution of other authors, contribution to development, validation and applications of LOTOS-EUROS

P. Builtjes: contributions to LOTOS development, applications of LOTOS-EUROS and description of model history

20 L. Curier: contribution to data assimilation applications

H. Denier van der Gon: contribution to emissions input

C. Hendriks: contribution to model validation, ammonia emissions, scenario simulations

S. Jonkers: contribution to model development and applications

R. Kranenburg: contribution to model development, labelling, deposition

25 J. Kuenen: contribution to emissions input

R. Timmermans: contribution to data assimilation, OSSE and operational forecasting

A. Segers: contribution to data assimilation and CAMS, model development, dust modelling

A. Visschedijk: contribution to emission input

A. van Pul: Contribution to development of EUROS, model applications

30 F. Sauter: contribution to model development of EUROS, LOTOS-EUROS and applications

E van der Swaluw: contribution to model development and applications

D. Swart :contribution to operational smog forecasting in the Netherlands

R. Wichink Kruit: contribution to model development (SIA and NH3)

H. Eskes: Contribution to data assimilation and forecasting



J. Douros: Contribution to model applications (CAMS)

E. van Meijgaard: Contribution to RACMO coupling and climate scenario simulations

B. van Ulft: Contribution to RACMO coupling and climate scenario simulations

P. van Velthoven: contribution to operational forecasting

5  A. Mues: Contribution to model applications (climate scenario and emission timing)

S. Banzhaf: Development of cloud chemistry and wet deposition approach, scenario analysis

R. Stern: Contribution to LOTOS development

G. Fu: Contribution to development of data assimilation system

S. Lu: Contribution of development of data assimilation system

10  A. Heemink; Contribution to development of  data assimilation system

N. van Velzen: Contribution to development of data assimilation system

M. Schaap: Major contributions to text, development and applications of LOTOS and LOTOS-EUROS



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





**Tables**

**Table 1. Input datasets used in LOTOS-EUROS model run for performance evaluation**

| Input | dataset |
|---|---|
| Domain | 5 vertical levels (5 km) <br> 15º W-35 º E, 35-70 º N, 0.5x0.25 º lonxlat |
| Land cover | Corine/Smiatek (EEA 2000) combined with <br> European Tree species data  Köble and  Seufert (2001) |
| Boundary conditions | Climatology +Mace Head correction |
| Meteorology | ECMWF 12 h forecasts |
| Wet Deposition description | Below-cloud scavenging coefficients (Scott 1978) |
| Dry deposition description | Restistance approach. Van Zanten et al 2010 for gases; Zhang et al (2001) for particles; compensation point approach for ammonia (Wichink Kruit et al 2010, 2012a) |
| Anthropogenic emissions | MACC-III |
| Biogenic emissions | Tree  species-dependent emission factors for isoprene (Schaap et al 2009, Beltman et al 2013) |
| Soil NOx emissions | Soil-temperature dependent (Novak& Pierce 1993) |
| Fire emission | MACC/CAMS GFAS product |
| Dust emissions | On-line calculation of natural dust; <br> agricultural activity; road resuspension |
| Sea spray emissions | On-line |
| Gas-phase chemistry | TNO CBM-IV |
| Secondary Inorganic Aerosol | Isorropia II (Fountoukis and Nenes, 2007) |
| Secondary Organic Aerosol | Not included |





**Table 2. Comparison of modelled and measured concentrations of air pollutants Bold = based on hourly measurements;** regular = based on daily measurements. Wet deposition is reported in monthly values. O₃ daymax and 8hmax are based on April-September. Concentrations in air are reported in µg/m3, concentrations in rain water in mg/l.

5  Notes: For SO₄ and Na, total aerosol matrix was taken (not $PM_{10}$) in order to have more stations available.

| Species | Mean correlation | Observed mean | Mean rmse | Mean bias | Measure for variability $\sigma_{obs}/\sigma_{mod}$ | Spatial correlation (Pearson) | #stations |
|---|---|---|---|---|---|---|---|
| **O3** | **0.61** | **60.4** | **21.7** | **4.78** | **1.09** | **0.68** | **52** |
| O3 daymax | 0.67 | 91.55 | 16.65 | 6.38 | 1.22 | 0.90 | 51 |
| O3 8hmax | 0.66 | 85.97 | 16.75 | 7.41 | 1.20 | 0.88 | 51 |
| **NO2** | **0.44** | **7.83** | **7.24** | **1.69** | **1.16** | **0.91** | **13** |
| NH3 | 0.26 | 1.70 | 1.65 | -0.015 | 1.47 | 0.95 | 13 |
| **SO2** | **0.33** | **1.14** | **1.67** | **0.33** | **1.13** | **0.79** | **9** |
| SO4 | 0.52 | 1.58 | 1.21 | -0.61 | 1.63 | 0.87 | 28 |
| NO3 | 0.50 | 2.08 | 2.04 | -0.15 | 1.05 | 0.92 | 15 |
| NH4 | 0.63 | 1.12 | 0.88 | -0.26 | 1.35 | 0.89 | 18 |
| EC | 0.67 | 1.06 | 0.82 | -0.26 | 2.42 | (1.0) | 2 |
| Na | 0.53 | 0.81 | 0.75 | 0.29 | 0.85 | 0.93 | 21 |
| PM25 | 0.54 | 9.57 | 7.12 | -3.11 | 1.65 | 0.88 | 15 |
| PM10 | 0.46 | 16.02 | 9.92 | -4.93 | 1.79 | 0.81 | 21 |
| TNH4 | 0.50 | 1.18 | 0.74 | -0.066 | 1.19 | 0.90 | 24 |
| wetNH4 | 0.51 | 0.54 | 0.41 | -0.20 | 2.61 | 0.71 | 56 |
| wetNO3 | 0.43 | 1.58 | 1.07 | -0.67 | 2.64 | 0.83 | 56 |
| wetSO4 | 0.28 | 1.18 | 0.95 | -0.78 | 5.43 | 0.69 | 56 |





**Figures**

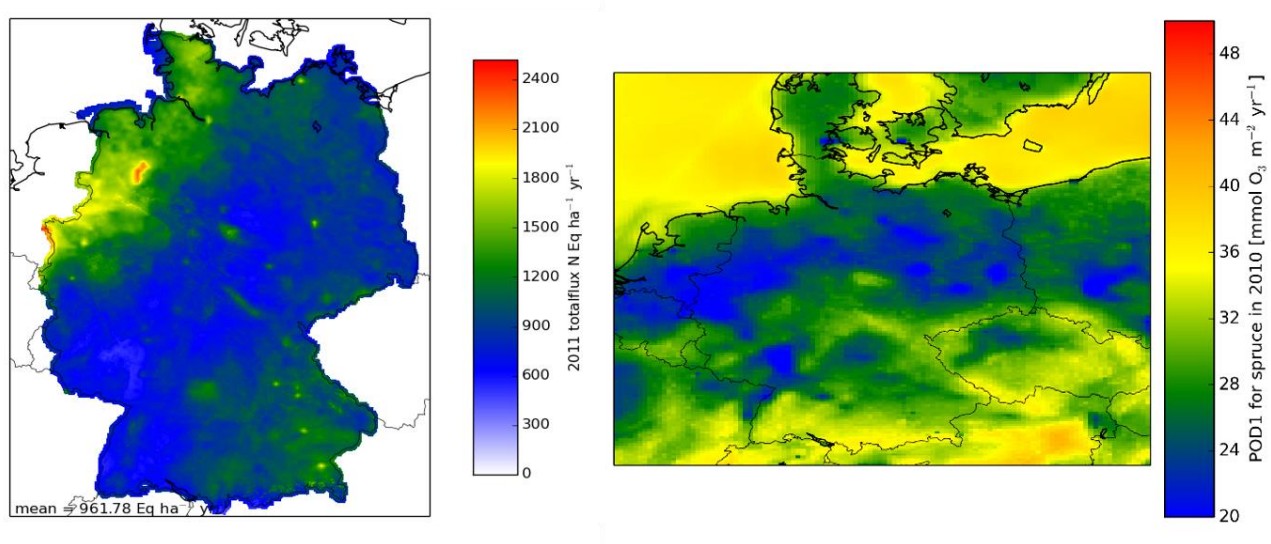

**Figure 1. Deposition of nitrogen (oxidised + reduced) over Germany, assessed using the LOTOS-EUROS model in**
5   **combination with observations of wet deposition (Schaap et al 2017), and phyto-toxic ozone dose for spruce (Bender et**
**al 2015).**





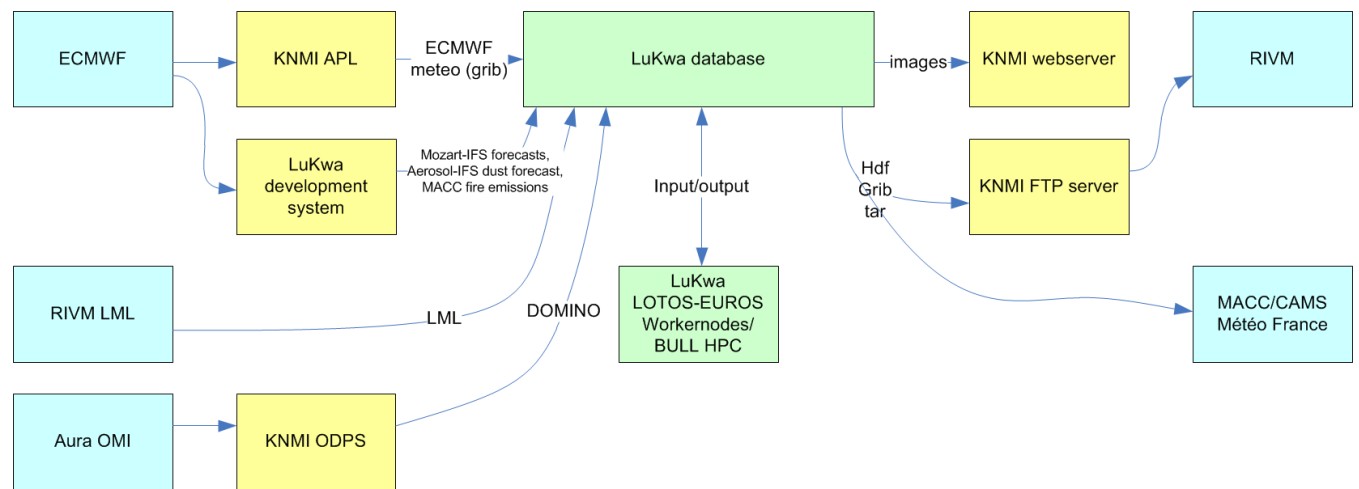

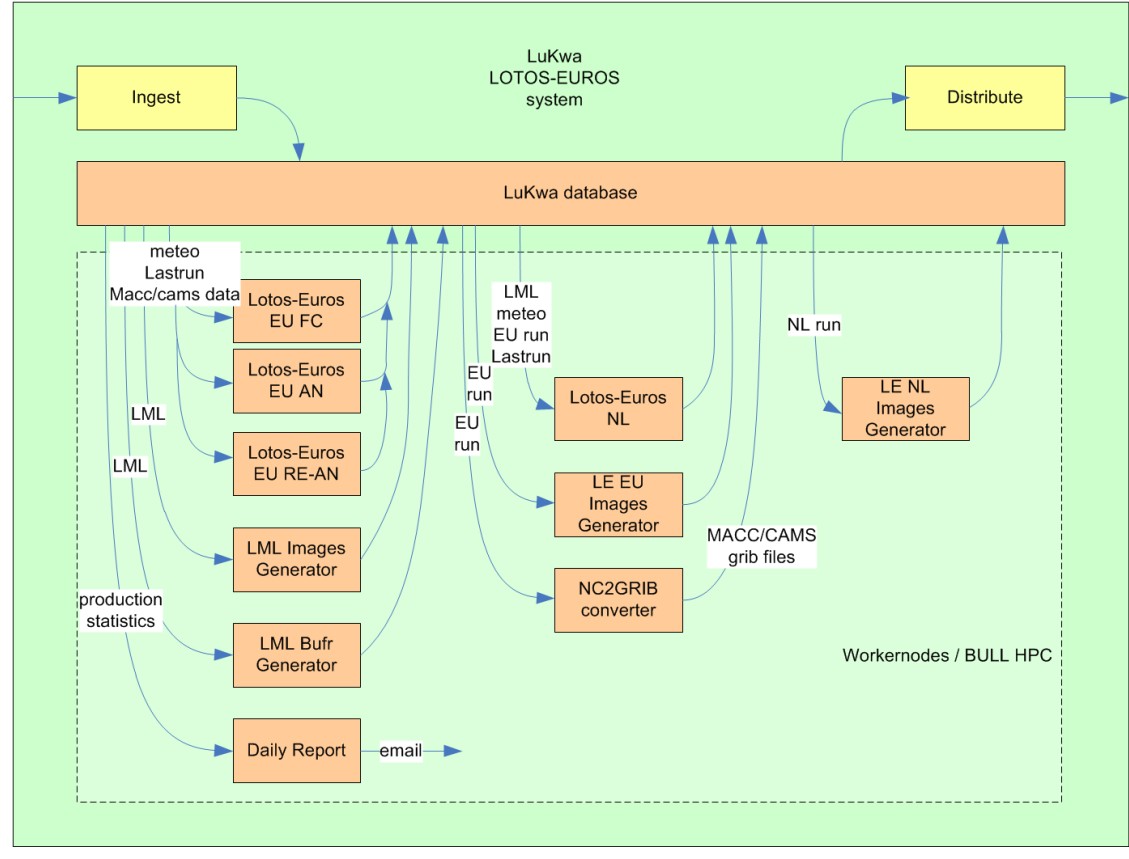

**Figure 2. Schematic view of operational suite at KNMI for the production of air quality analysis (AN) and forecasts (FC) with LE (LOTOS-EUROS) for the Netherlands (NL) and Europe (EU). In data assimilations (Dutch) ground observations of ozone are included (LML) and the European-scale simulation serves as boundary condition for the higher-resolution simulation for the Netherlands.**



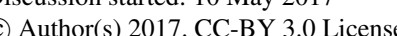

**Figure 3. Annual mean surface concentrations for several model species. Note the difference in scaling between the different species and the nonlinear scaling for several species.**



**Figure 3 (Continued)**





**Figure 4. Time series of observed and modelled concentrations of ozone and PM$_{10}$ at Vredepeel (the Netherlands, 5.85 E, 51.54 N) and concentrations of NO$_3$ in rain water in Kollumerwaard (the Netherlands, 6.28N, 53.33 N) .**



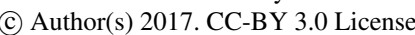

Figure 5. Modelled dry (left) and wet (right) deposition of reduced (top) nitrogen, oxidised (middle) and oxidised sulphur (bottom)



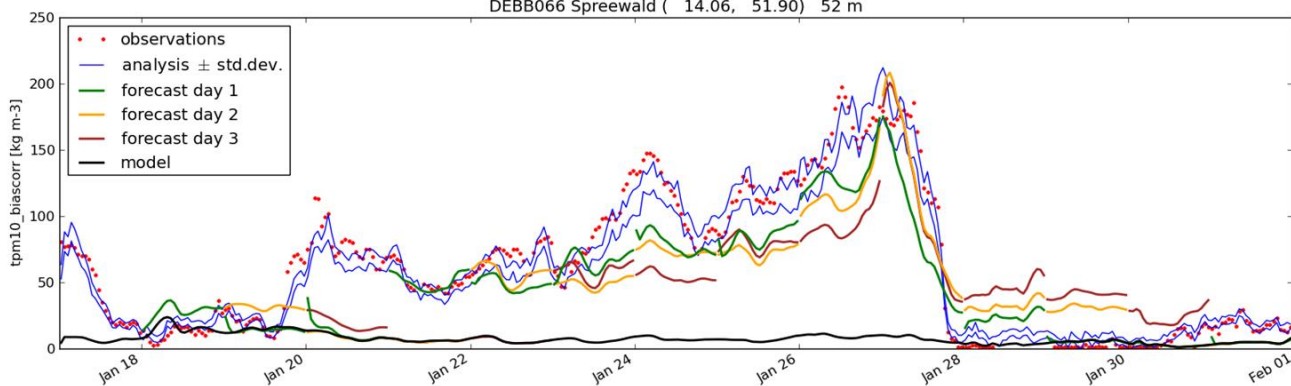

**Figure 6. Impact of ground-based PM$_{10}$ data assimilation on analysis and three-day forecast during a large-scale episode of high PM$_{10}$ concentrations over large parts of Europe.**

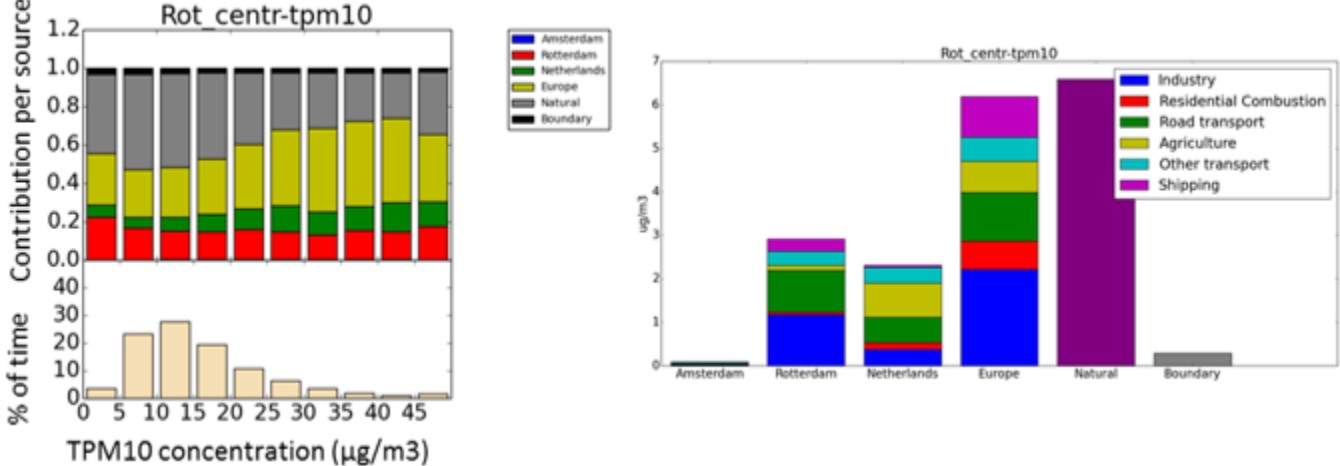

5 **Figure 7. Source apportionment for Rotterdam City Centre, 2011. Relative contribution of several source sectors to daily average PM$_{10}$ concentrations (left) and annual average absolute contributions of PM$_{10}$ from several source regions and sectors (right).**





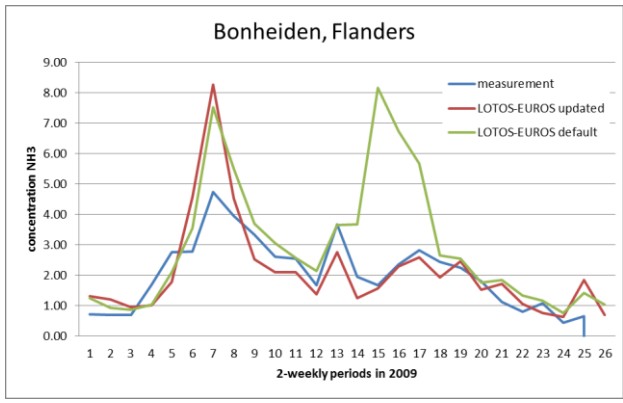

**Figure 8. Time series of modelled and measured ammonia concentrations in Bonheiden (Belgium) in 2009, showing the reduction of the unrealistic second NH3 peak of the default simulation around week 16.**

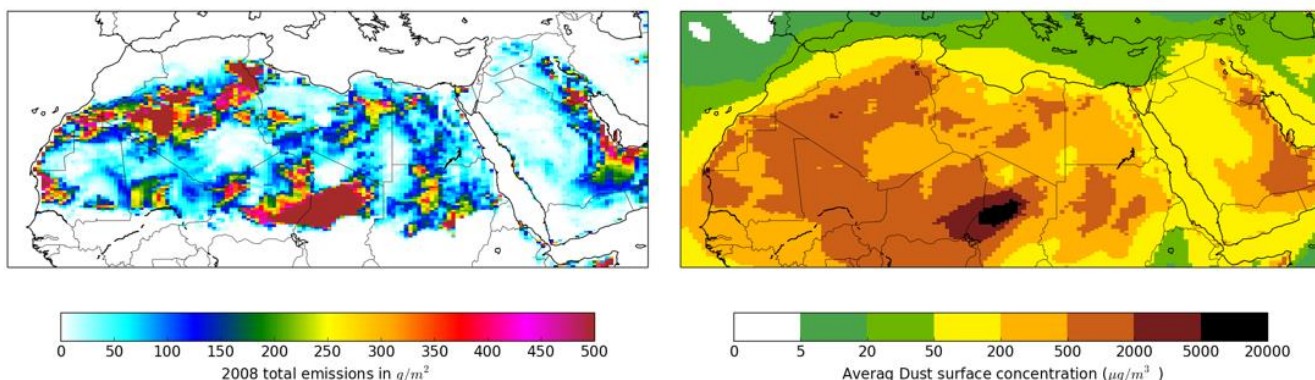

5 **Figure 9. Annual total dust emissions and annual average dust concentration for 2008.**

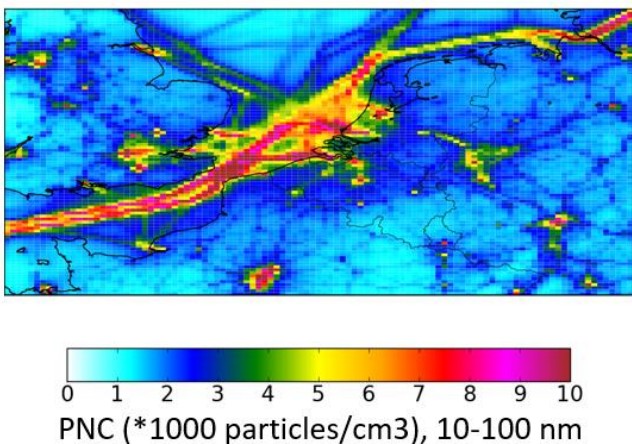

**Figure 10. High-resolution modelling of particle number concentrations in the ultrafine range (10-100nm), annual mean value for 2009.**