# Peer review of "Curriculum Vitae of the LOTOS-EUROS (v2.0) chemistry transport model"

_Geoscientific Model Development, 2017_

## Referee Comment (RC1) · Anonymous Referee #1 · 9 Jun 2017

Review: 'Curriculum Vitae of the LOTOS-EUROS (v2.0) chemistry transport model' by Astrid M. M. Manders, et al. in GMD, manuscript number: gmd-2017-88

This paper presents the latest version of the LOTOS-EUROS model (v2.0) which is open source. It outlines typical model settings/conditions, as well as optional features that are under development. It also provides an historical perspective on the model development, and provides a substantial number of references for its application and improvements to the model performance. Overall I think this paper should be published, however there are a few revisions that I would recommend before publication. For one, there are a number of English language edits that should be taken care of that would improve the readability of the paper.

Comments:

[Figure]

P4L1-2: Is this really a contrast? The US is also one country unless North America is being referred to. And in that sense, it was one community in the same way that each country in Europe has a community.

P4L14-17: These lines seem to be in direct disagreement with what this paper is doing. While the authors may argue that the paper is different from other model description papers, that is also basically what this paper is as well, even if there is some other information included. I would suggest removing this as it seems to be a contradiction.

P5: why is it that the LOTOS model development is described in detail whereas the EUROS model gets references. Even if it has been written about before, this is very unbalanced, it would be good to include some short summary for EUROS as well, so that the sections are a bit more similar.

P6L6: Why does this sentence say 2004 when all previous text says 2005?

P6L9: This section would be more appropriately titled if it included something like 'development and applications' in the subject heading.

P6L17: What 'new EU legislation' is being referred to? It would be good to include a year, also because PM10 and PM2.5 regulations came into effect at different times. Also what is meant by the 'new European monitoring capacity'?

P7: the paragraph that ends at the top of the page and the first full paragraph need a connecting sentence. The earlier paragraph ends as if the PM discussion were closed and then the authors go on to discuss nitrogen and ammonia emissions. I would suggest something along the lines of 'Another aspect of modeling PM that has received significant attention is secondary inorganic aerosol.'

P7L15: what is the 'compensation point for ammonia'? Please explain the concept as it is referred to multiple times throughout the manuscript. (also on P9L8)

P8L1: include a very short (less than a sentence) description of what AQMEII-2 is.

P8L2-3: this sentence seems to be a bit of a non-sequitur to the sentences surrounding it.

P8L5-6: ref??

P8L8-10: This sentence is not clear at all.

P8L24: this doesn't make sense. What does it mean to 'warrant an availability' and what does that have to do with KNMI?

P9L5-6: that the VBS approach depends heavily on assumptions – isn't that true for many things in modelling? Be more specific rather than making such a vague statement. Why or what aspects make this more dependent on assumptions than any of the other inputs or modules? If it doesn't fit here, reference the discussion elsewhere.

P9L6-8: explain this, not clear currently. Level of detail of a process should match the general level of accuracy of a model????

Section 4: I assume that the description that is included about the features of the model includes the options used for the model evaluation/plots presented in this work, as well as additional options that could be used. It would be good to be more explicit about this here. The reason that I mention this is because in e.g., emissions, the TNO-MACC-III database is discussed as being used as input, but other EI can be used as well. So this isn't really a model feature.

P11L8-10: the description of the layers needs to be clarified. There are a lot of different layers mention that are all relatively close to the surface. Do any of these overlap? How do they fit together? It might be good to include a depiction of these.

P11L24-25: the tracers can be easily selected. Does this refer only to CO2 or also other tracers? Please clarify.

P12L1: is the heterogeneous chemistry on wet aerosol also part of IsorropiaII or is that a different module? Also, the cloud chemistry, no module is mentioned, is this part of

the core model or ?

P12L5-7: so SOA isn't really taken into account, is anything used to compensate or otherwise account for SOA? Please address this.

P12L15-16: The authors mention that when not all met fields are available, representative average values can be used. Are there a minimum number of or certain critical parameters where an average would not be acceptable? E.g., for these certain parameters, rep avg values are acceptable?

P12L28: will people know what the 'sand blasting approach' is? Might be good to describe this a bit.

P12L30: 'region-specific' and 'local' seem redundant/one would supersede the other. Prob best to choose just one, or be more explicit.

P13L1-4: the TNO MACC III databse was used for emissions. In this study for the model performance runs? In all of the papers presented? The context needs to be more explicitly clear. Also, why PPM and not PM?

P13L6-7: the temporal disaggregation that is mentioned, are these defined by the user? As part of the EI? If these are model defined, it would be good to include a table or similar, as in e.g., the Simpson paper on the EMEP model, to document what these are. Can be in the SI.

P13L27-29: Why is the reference for these processes that is included in Table 1 not included here? Also, it is not clear whether one approach needs to be chosen or if both are used in parallel given the following sentences.

P13L32-P14L1: this aspect of translating concentrations to obs height needs more explanation. It is not clear at all.

P14L22-25: if the effect of these is so large, what makes what is chosen the best? What should be considered?

P15L4-5: what does this sentence mean? What is being distinguished here with the aerosol samplers and PM10 samplers?

P15L27-28: why are dust and sea salt boundary conditions not used?

P15L32: explicitly mention what kind of emissions are coming from the Ruhr area as not everyone might know this.

P16:1-6: Couldn't this underestimate also be related to the lack of SOA in the model? How is this accounted for?

P16L8-13: Spatial correlation and spatial correlation for annual mean are both referred to, with one yielding good results and the other not, in addition Table 2 also has two columns (mean and spatial correlation). Please outline what the differences in these two different correlations are, with explanations as to why the differences exist, as it is not clear from the current text.

P16L26-27: How are Table 2 and Figure 3 showing a comparison with rain water conc observations? Rain water doesn't seem to be among any of the stats listed or plotted, unless this is a ref to the e.g., wetNH3 in table 2.

P16L31: as above, explain what exactly is being correlated.

P17L5 (also P21L13): technically, aerosol includes the gas-phase. I would aim to be consistent and refer to gas-phase and PM.

P17section6: I would suggest mentioning already how to get access to these features that are not part of the open source version, or where to find this information. It would make it more user/reader-friendly.

P17L11-12: please mention explicitly what the aim of these techniques are, so that the reader easily understands what is being improved through their application.

P17L24-26: as with the previous comment, please explain what this technique does – how is it different from all those already mentioned above?

P18L1/Figure 6: is the black line the model w/o assimilation? If so, can the model output really be all that useful w/o incorporating assimilation techniques? Or was this example chosen because of the significant improvement?

P18L8-9: does the reference apply to both AOD and NO2 or just AOD? If both, please move the reference, if only AOD, ok, but then please add a reference for NO2 if possible.

P19L8: 'leads to better model performance' – can some more quantitative description be given for the improvement?

P21L2-3: this sentence is not clear and doe3sn't fit together. Need to extend the community and therefore the code was made open source? Or ?

P21L19-20: how is the improvement of the emissions one of the focal points of LOTOS-EUROS? These are two separate products, and models can be used to test changes to EI, but are model developers working on this? This doesn't really seem to fit?

P21L29-P22L6: all solid points, but it seems to be a bit of a laundry list. How realistic are these different points? Are they all equally close to realization? And even if all of these were implemented, would the model then even be able to run anymore in an acceptable amount of time? It would be good to have a somewhat more nuanced discussion of such points rather than just listing them off.

P22L13-14: the description of the vertical structure and the link to the horizontal grid is not at all clear, nor what the implications are. Also on L17, the layers of the driving met model??? Please clarify.

Figure 7: why is 'natural' all from 'shipping'? That does not line up.

Minor edits:

-there seem to be a lot of spacing issues where there are multiple spaces instead of 1. Make sure this is corrected at the proof stage.

-various subscripts in chemical shorthand for e.g. NH4 are missing. Please find and correct.

P2L2: 'combining' rather than 'combination of'

P2L13: no comma after the ()

P2L18: '...to keep a good...'

P2L19: '...to keep a good record of the effect...'

P3L16: ) missing at end of reference

P3L23: 2nd reference needs an e.g.

P3L24: include the abbreviation CTMs here, as it needs to be defined before the abbreviation is used later; also in this line replace 'nowadays' with 'currently'

P3L29: '...(e.g. Baklanov et al., 2014), as well as informing the design of monitoring strategies....'

P4L1: '...were developed...'

P5L4: remove 'have'

P5L22: remove 'has'

P5L23: write out 'RCG' or explain

P5L23-24: '....sister models with intensive exchange among their developers during their development.' (or something similar, as the models themselves cannot exchange knowledge, only the people behind the models)

P5L29: no apostrophe in POPs. That indicates possession. There is an extra ( in the Jacobs and van Pul ref.

P6L11: '... specific feature that uses a dynamic ...'

P6L12: '. . .enabling the application of the model. . .'

P6L14: '. . ..and advances in remote . . ..'

P6L22: ) needed after 2012

P6L25: in the references, either a ref is missing, or the ; is in error

P6L31: 'To better understand the origin of PM, a . . ..'

P6L32: '. . .which enables quantification of the . . ..'

P7L18: remove 'also'

P7L21: '. . .improve the model's skill to capture the intra-annual . . .'

P7L24: ozone is not capitalized

P7L28: remove 'to perform' and add at the end 'to be run' or 'to be performed'

P7L29: '. . .to evaluate scenarios including . . ..' The following list is confusing, is it climate change and energy policies, air quality mitigation, and land use change. Or separate climate change futures, energy policies, etc? Please clarify.

P7L31/L34: be consistent with abbreviation for RACMO-2

P7L31: explain transient scenario simulations.

P8L16: '. . .project was preparation for the . . .'

P8L17: '. . . forecasts made it possible to use . . .'

P8L22: a.o. ??

P9L15: 'adds'

P9L28: remove 'also'; '. . .test was updated to cover 2012 as well.'; Make sure the cases in your sentences and paragraphs match. This is not the only occurrence of this.

P9L10: explicitly mention that 2003 was the heatwave

P9L29: '. . . in a new way to investigate behavior. . .'

P9L31: something seems to be missing in this sentence. As compared to what??

P11L20: was aimed at?? Is still primarily aimed at??

P11L30: remove 'the'

P12L5: '. . .is not currently taken. . .'

P12L11: replace 'also' with 'and'

P12L13: 'The default for the model is 3-hourly ECMWF. . .'

P12L15: 'online' is one word

P12L22: do you mean 'variables' when you use 'relations'? or relationships? Relations is not correct.

P12L25: 'are' not 'was'

P13L8: 'in the vertical, fixed emission profiles. . .'

P13L9: 'If desired, scenario factors for specific . . .. can be integrated without changes to the code.'

P13L19: 1/112th as a decimal for consistency.

P13L23-24: '. . .database, were relabeled as "ocean or sea", since. . .'

P13L27: replace 'over' with 'between'; '. . .and below-cloud scavenging.'

P14L13-14: move 'also' to '. . . data are also available. . .'

P14L22-23: move 'often' to '. . .models are often used. . .'

Section 5.2: consistency in case – sometimes past, sometimes present, edit this

P15L7: '.. all data points with . . .'

P15L14: write out RMSE the first time it is used

P15L31: 'Figure 4 illustrates a time series for ozone and PM10 for a station in the Netherlands, comparing observed and modeled concentrations.'

-There are a lot of other edits that should be taken care of for English, please have a native English speaker read through the paper to catch these things.

P19L31/P20L10: why are these words bolded?

P20L17: the BL is part of the troposphere, please edit wording to make this clear.

P20L16: include a reference for the original nucleation scheme by Vehkamäki

P21L16: 'the role of reactive nitrogen becomes more clear' please rephrase. How so?

P21L22-23: how are met conditions going to be modeled more realistically in the future? Do the authors mean that 'by taking met conditions into account', the soil Nox, etc will be modeled more accurately?

Table 1: for anthro emissions, please be consistent with the EI abbreviation. Previously it was TNO-MACC-III; for fire emission, is this the same as the Kaiser ref? please add it if so.

Figure 4: please add labels to each of the plots (a, b, c) and update the references to them in the text. In the legend, an explanation for the red dots is missing.

---

## Referee Comment (RC2) · Dr Rouil (Referee) · 20 Jul 2017

The idea of a curriculum vitae for the LOTOS-EUROS model, which is one of the most famous and efficient chemistry-transport model in Europe is very interesting and welcome in this period when, as mentioned by the authors such models have reached a certain level of maturity. The article also introduces some history about the development of air quality models over the twenty past years which is also well documented. As a general comment, we have here a good and relevant paper. The main added-value of this paper is to explain the history of a chemistry-transport model like LOTOS-EUROS, the drivers that led to the current open-source version, the current and future challenges. In that perspective it would have been interesting to get more details about the merging process between both initial models LOTOS and EUROS. EUROS is briefly

described compared to LOTOS and how the Dutch teams managed to get only one model is not really explained although this is a quite original approach. For instance, it would be relevant to know how the experts selected the model parametrizations, etc ... (in paragraph 2.1).

I understand the authors decided to tell a story avoiding equations describing more in detail the model (this is not a peer-reviewed model description). This choice is valuable to focus on the model development strategy, but as a consequence, some parts of the paper may remain a little unclear for non experts readers (reference to EnKF method and VBS model in paragraphs 6.1 and 6.4 for instance or the paragraph on emissions modelling -6.3- which may be confusing for someone who did not understand that emissions modelling is a part of the CTM). Also, the discussion on high resolution runs (page 22) is relevant but not conclusive (finally is the combination of LOTOS-EUROS and OPS the most promising approach? addedd avlue of the plume-in-grid?) I would recommend to review those paragraphs and to amend them with few more explanations.

This paper is pleasant to read for air quality modellers who share the authors' concerns and philosophy but may be more difficult for people outside the field. It is mainly due to the fact that a number of references and definitions, useful for good understanding, is missing . I have noted the following terms used without any explanations: GEOSS, PoDY, Ensemble approach, NOy, SDS-WAS and some references to models : HARMONIE, COSMO, VBS, OPS... few words to introduce them are necessary.

Please not that schemes provided on figure 2 are very difficult to understand, and for this reason are almost useless. It is necessary to revise and simplify them, making the acronyms more explicit and focusing on the key messages those schemes are supposed to bear. The other figures are correctly chosen. I would just recommend to add, if possible, a representation of modelling fields or their performances with assimilation of satellite information. This aspect is well discussed in the paper but not illustrated.

A good point is the extensive bibliography provided. However, I would recommend to add a reference (for example page 4 line 15 with Menut et al. 2013) to the last peer-reviewed paper related to the evolutions of the CHIMERE model (https://www.geosci-model-dev.net/10/2397/2017/).

Here are a number of typo corrections I noted: - Page3 line 16 a parenthesis is missing after 1972; - Page 16 line 31 "months" instead of "mont", - Page 21 line 26 a dot is missing - Page 22 line 17, add a "s" to "application", - Page 22 line 18 "from" instead of "form", line 19 add a reference for HARMONIE and COSMO (at least institutes that develop them), - A number of times in the text PM10, PM2,5, NO2 or NOX are written without indices - Check the References to "Collette" , the correct name is "Colette"

---

## Author Comment (AC1) · 5 Sep 2017

Review: `Curriculum Vitae of the LOTOS-EUROS (v2.0) chemistry transport model' by Astrid M. M. Manders, et al. in GMD, manuscript number: gmd-2017-88 This paper presents the latest version of the LOTOS-EUROS model (v2.0) which is open source. It outlines typical model settings/conditions, as well as optional features that are under development. It also provides an historical perspective on the model development, and provides a substantial number of references for its application and improvements to the model performance. Overall I think this paper should be published, however there are a few revisions that I would recommend before publication. For one, there are a number of English language edits that should be taken care of that would improve the readability of the paper.

*We thank the referee for the conscious and critical reading and detailed comment, which has contributed significantly to the quality of the text of the revised paper. The issues that were raised are answered in detail below in italics.*

Comments:

P4L1-2: Is this really a contrast? The US is also one country unless North America is being referred to. And in that sense, it was one community in the same way that each country in Europe has a community.

*What we wanted to indicate here is that in North America only a few models exist but with rather large communities, whereas in Europe many models were developed, with each their own small communities. This has set the scene for the model intercomparison studies on Europe, which do not have an American counterpart before AQMEII to our knowldge .Text has been adapted.*

P4L14-17: These lines seem to be in direct disagreement with what this paper is doing. While the authors may argue that the paper is different from other model description papers, that is also basically what this paper is as well, even if there is some other information included. I would suggest removing this as it seems to be a contradiction.

*We understand the point of the reviewer. Indeed, our paper does have many aspects in common with other model development and description papers, like references to parameterizations and an evaluation. Other model papers often include detailed descriptions of their parameterizations, including formulas, which we wanted to avoid here because these are already covered in the reference guide. What we wanted to achieve is to put model development more in context and give a broader overview that what we see is generally done. The text is modified to reflect this point.*

P5: why is it that the LOTOS model development is described in detail whereas the EUROS model gets references. Even if it has been written about before, this is very unbalanced, it would be good to include some short summary for EUROS as well, so that the sections are a bit more similar.

*We agree that this is unbalanced but it is difficult to find this information for EUROS. We did not manage to get input from the people involved in the early development of EUROS.*

P6L6: Why does this sentence say 2004 when all previous text says 2005?

*The process was started in 2004 and completed in 2005, in 2005 the version was released. This was added to the text.*

P6L9: This section would be more appropriately titled if it included something like 'development and applications' in the subject heading.
*The section title has been adapted*

P6L17: What 'new EU legislation' is being referred to? It would be good to include a year, also because PM10 and PM2.5 regulations came into effect at different times.
Also what is meant by the 'new European monitoring capacity'?

*Reference to EC 99 and EC 2008 added, and sentence changed (The European monitoring network and methods related to this legislation for PM10, and later on also PM2.5, clearly revealed a systematic gap between observed and modelled concentrations)*

P7: the paragraph that ends at the top of the page and the first full paragraph need a connecting sentence. The earlier paragraph ends as if the PM discussion were closed and then the authors go on to discuss nitrogen and ammonia emissions. I would suggest something along the lines of 'Another aspect of modeling PM that has received significant attention is secondary inorganic aerosol.'

*The sentence 'Secondary inorganic aerosols have also gained significant attention in the light of reactive nitrogen.'has been added*

P7L15: what is the 'compensation point for ammonia'? Please explain the concept as it is referred to multiple times throughout the manuscript. (also on P9L8)
*Clarification added: ' the compensation point for ammonia, which describes the net deposition velocity taking into account ammonia re-emissions from the surface…'*

P8L1: include a very short (less than a sentence) description of what AQMEII-2 is.
*Also a semi-online coupling between RACMO2 and LOTOS-EUROS was established (Baklanov et al 2014) and contributed simulations to Phase II of the Air Quality Model Evaluation International Initiative (AQMEII )(Brunner et al 2014, Im et al 2015a,b) in which on-line coupled models were evaluated for simulations of ozone and particulate matter for the year 2010. .*

P8L2-3: this sentence seems to be a bit of a non-sequitur to the sentences surrounding it.
*Sentence has been added to better guide the reader:Also land use change scenarios have been explored with LOTOS-EUROS.*

P8L5-6: ref??
*The reference seems all right, we do not understand the issue*

P8L8-10: This sentence is not clear at all.
*Rephrased: A study to the impact of shifting from combustion energy production to solar and wind energy without ample energy storage showed that air quality is less positively affected than is often assumed, since combustion energy is needed to replace solar and wind energy at instances where air pollution accumulation is favoured*

P8L24: this doesn't make sense. What does it mean to 'warrant an availability' and what does that have to do with KNMI?
*Rephrased: The LOTOS-EUROS forecasting service is run at the Royal Dutch Meteorological Institute (KNMI) in an operational weather forecasting environment so that disturbances are quickly resolved, warranting….*

P9L5-6: that the VBS approach depends heavily on assumptions – isn't that true for many things in modelling? Be more specific rather than making such a vague statement.
Why or what aspects make this more dependent on assumptions than any of the other inputs or modules? If it doesn't fit here, reference the discussion elsewhere.
*A reference to section 4.3 and 6.4 is made, and the description in 4.3 has been extended.*

P9L6-8: explain this, not clear currently. Level of detail of a process should match the general level of accuracy of a model????
*It doesn't make sense to model one process in great detail if uncertainties or inaccuracies in other processes , inputs and model resolution would counteract the net effect of such detailing. The sentence has been adapted.*

Section 4: I assume that the description that is included about the features of the model includes the options used for the model evaluation/plots presented in this work, as well as additional options that could be used. It would be good to be more explicit about this here. The reason that I mention this is because in e.g., emissions, the TNO-MACC-III database is discussed as being used as input, but other EI can be used as well. So this isn't really a model feature.
*We agree with the reviewer that in particular for anthropogenic emissions the description is too narrow. The text has been extended to included other inventories as well.*

P11L8-10: the description of the layers needs to be clarified. There are a lot of different layers mention that are all relatively close to the surface. Do any of these overlap? How do they fit together? It might be good to include a depiction of these.

*We did not include a picture of the model structure but we changed the text to make the description more explicit.*

P11L24-25: the tracers can be easily selected. Does this refer only to CO2 or also other tracers? Please clarify.
*Clarifying examples have been added*

P12L1: is the heterogeneous chemistry on wet aerosol also part of IsorropiaII or is that a different module? Also, the cloud chemistry, no module is mentioned, is this part of the core model or ?
*heterogeneous chemistry is not part of the ISORROPIA II routine or gas-phase chemistry but is a separate module, using the aerosol water as calculated by ISORROPIA II. Cloud chemistry was not present in v1.0 but is now part of the core model. This has been clarified in the text.*

P12L5-7: so SOA isn't really taken into account, is anything used to compensate or otherwise account for SOA? Please address this.
*No, there is no compensation for such missing component. The motivation for not including SOA has been added explicitly to the text.*

P12L15-16: The authors mention that when not all met fields are available, representative average values can be used. Are there a minimum number of or certain critical parameters where an average would not be acceptable? E.g., for these certain parameters, rep avg values are acceptable?

*The statement is indeed too strong/general. This approach makes sense for compartments with buffer capacity: soil properties, as in the example, or sea water temperature, but not for parameters that should represent synoptic timescales. The statement has been rephrased accordingly.*

P12L28: will people know what the 'sand blasting approach' is? Might be good to describe this a bit.
*The process of dust emissions has been extended to better clarify its nature and which processes are included.*

P12L30: 'region-specific' and 'local' seem redundant/one would supersede the other. Prob best to choose just one, or be more explicit.
*Region-specific has been deleted*

P13L1-4: the TNO MACC III databse was used for emissions. In this study for the model performance runs? In all of the papers presented? The context needs to be more explicitly clear. Also, why PPM and not PM?
*Text has been adapted to clarify this. PPM stands for primary particulate matter, this has been stated explicitly in the text now.*

P13L6-7: the temporal disaggregation that is mentioned, are these defined by the user? As part of the EI? If these are model defined, it would be good to include a table or similar, as in e.g., the Simpson paper on the EMEP model, to document what these are. Can be in the SI.
*The text has been extended to clarify these points, the original reference is included.*

P13L27-29: Why is the reference for these processes that is included in Table 1 not included here? Also, it is not clear whether one approach needs to be chosen or if both are used in parallel given the following sentences.
*The reference in Table 1 is now added to the main text. The text has been clarified, one has to choose between the two options.*

P13L32-P14L1: this aspect of translating concentrations to obs height needs more explanation. It is not clear at all.
*The idea of the approach has been stated more explicitly*

P14L22-25: if the effect of these is so large, what makes what is chosen the best? What should be considered?
*The motivation has been added, including a reference to an analysis of the MACC-reanalysis product.*

P15L4-5: what does this sentence mean? What is being distinguished here with the aerosol samplers and PM10 samplers?
*Some aerosol samplers do not have an explicit inlet specification but are reported as _aerosol, these data are included for SIA species to increase the number of available stations. This is now stated in the text.*

P15L27-28: why are dust and sea salt boundary conditions not used?
*An explanation has been added to 5.1*

P15L32: explicitly mention what kind of emissions are coming from the Ruhr area as not everyone might know this.
*Added to the text*

P16:1-6: Couldn't this underestimate also be related to the lack of SOA in the model? How is this accounted for?
*The reviewer is right, also SOA will be part of the underestimation. This point is now addressed in the text.*

P16L8-13: Spatial correlation and spatial correlation for annual mean are both referred to, with one yielding good results and the other not, in addition Table 2 also has two columns (mean and spatial correlation). Please outline what the differences in these two different correlations are, with explanations as to why the differences exist, as it is not clear from the current text.
*Clarifications were added to main text and table caption.*

P16L26-27: How are Table 2 and Figure 3 showing a comparison with rain water conc observations? Rain water doesn't seem to be among any of the stats listed or plotted, unless this is a ref to the e.g., wetNH3 in table 2.
*The reference to figure 3 is indeed the wrong one here. The caption of Table 2 and the main text have been adapted to clarify this issue, we indeed refer to wetNH4, wetNO3 and wetSO4, and the behavior is illustrated in Figure 4. Figure 5 illustrates the annual totals of wet and dry deposition, without reference to rainwater concentration.*

P16L31: as above, explain what exactly is being correlated.
*Explained in the text*

P17L5 (also P21L13): technically, aerosol includes the gas-phase. I would aim to be consistent and refer to gas-phase and PM.
*Done*

P17section6: I would suggest mentioning already how to get access to these features that are not part of the open source version, or where to find this information. It would make it more user/reader-friendly.
*Although this information is already at the end of the paper we included it here as well, and we tried to make a better separation between what is included and excluded.*

P17L11-12: please mention explicitly what the aim of these techniques are, so that the reader easily understands what is being improved through their application.
*We assume that the referee refers to L12-13.We indicated the aim more clearly.*

P17L24-26: as with the previous comment, please explain what this technique does – how is it different from all those already mentioned above?
*Explanatory sentence was added.*

P18L1/Figure 6: is the black line the model w/o assimilation? If so, can the model output really be all that useful w/o incorporating assimilation techniques? Or was this example chosen because of the significant improvement?

*This example indeed deserves more attention. This example is chosen because it represented an episode with exceptionally high PM10 concentrations over large parts of Europe, that could not be explained by merely changing the emissions of the existing tracers. For the baseline concentrations the model performs reasonably well, but for the episode with high concentrations an unspecified tracer had to be added. This tracer could represent large contributions of SOA from wood burning or wind-blown dust from bare soil during this cold and dry period. The example shows the considerable gain in forecast skill by using data assimilation in the right way. This explanation is added to the main text.*

P18L8-9: does the reference apply to both AOD and NO2 or just AOD? If both, please move the reference, if only AOD, ok, but then please add a reference for NO2 if possible.
*Reference to ISOTROP final report added for NO2.*

P19L8: 'leads to better model performance' – can some more quantitative description be given for the improvement?

*Yes, the paper by Mues et al (2013) provides relevant values, that are now added to the text: When using all new time profiles simultaneously the time correlation coefficient of daily average values increased by 0.05 (NO2), 0.07 (SO2) and 0.03 (PM10) at urban background stations in Germany.*

P21L2-3: this sentence is not clear and doesn't fit together. Need to extend the community and therefore the code was made open source? Or ?
*This sentence was moved to the beginning of the paragraph and slightly modified to be more clear.*

P21L19-20: how is the improvement of the emissions one of the focal points of LOTOSEUROS? These are two separate products, and models can be used to test changes
to EI, but are model developers working on this? This doesn't really seem to fit?

*It is not so straightforward to separate emissions and air quality modeling in practice. The sentence has been altered to read : "is a focal point for development and applications of LOTOS-EUROS". Data assimilation with LOTOS-EUROS can indicate structural changes in emission inventories and emission time profiles, and more accurate emissions are important for good model performance, in particular when higher resolutions are used. Model developers are researchers who spend part of their time to activities like improving time distributions of anthropogenic or natural emissions (e.g. Mues et al 2014, Hendriks et al 2016b ), which is work of different nature that constructing annual total emission inventories.The text has been adapted to reflect this.*

P21L29-P22L6: all solid points, but it seems to be a bit of a laundry list. How realistic are these different points? Are they all equally close to realization? And even if all of these were implemented, would the model then even be able to run anymore in an acceptable amount of time? It would be good to have a somewhat more nuanced discussion of such points rather than just listing them off.
*The paragraph has been extended to indicate how difficult/relevant it would be to realize the points.*

P22L13-14: the description of the vertical structure and the link to the horizontal grid is
not at all clear, nor what the implications are. Also on L17, the layers of the driving met
model??? Please clarify.
*The text has been made more explicit.*

Figure 7: why is 'natural' all from 'shipping'? That does not line up.
*'Natural' is a separate bar, with by nature only natural contribution. It is  in a different colour scale than shipping but we understand the confusion, both being purplish. The choice of colours is adapted and the caption is extended to clarify this.*

Minor edits:
*The text has been checked for consistency of cases, spacings, subscript, and several minor corrections have been made including the suggested corrections.*

-there seem to be a lot of spacing issues where there are multiple spaces instead of 1. Make sure this is corrected at the proof stage.
-various subscripts in chemical shorthand for e.g. NH4 are missing. Please find and
correct.
P2L2: 'combining' rather than 'combination of'
P2L13: no comma after the ()
P2L18: '. . .to keep a good. . .'
P2L19: '. . .to keep a good record of the effect. . .'
P3L16: ) missing at end of reference
P3L23: 2nd reference needs an e.g.
P3L24: include the abbreviation CTMs here, as it needs to be defined before the abbreviation
is used later; also in this line replace 'nowadays' with 'currently'
P3L29: '. . .(e.g. Baklanov et al., 2014), as well as informing the design of monitoring
strategies. . ..'
P4L1: '. . .were developed. . .'
P5L4: remove 'have'
P5L22: remove 'has'
P5L23: write out 'RCG' or explain
P5L23-24: '. . ..sister models with intensive exchange among their developers during

their development.' (or something similar, as the models themselves cannot exchange knowledge, only the people behind the models)

P5L29: no apostrophe in POPs. That indicates possession. There is an extra ( in the Jacobs and van Pul ref.

P6L11: '. . . specific feature that uses a dynamic . . .'

P6L12: '. . .enabling the application of the model. . .'

P6L14: '. . ..and advances in remote . . ..'

P6L22: ) needed after 2012

P6L25: in the references, either a ref is missing, or the ; is in error

P6L31: 'To better understand the origin of PM, a . . ..'

P6L32: '. . .which enables quantification of the . . ..'

P7L18: remove 'also'

P7L21: '. . .improve the model's skill to capture the intra-annual . . .'

P7L24: ozone is not capitalized

P7L28: remove 'to perform' and add at the end 'to be run' or 'to be performed'

P7L29: '. . .to evaluate scenarios including . . ..' The following list is confusing, is it climate change and energy policies, air quality mitigation, and land use change. Or separate climate change futures, energy policies, etc? Please clarify.

P7L31/L34: be consistent with abbreviation for RACMO-2

P7L31: explain transient scenario simulations.

P8L16: '. . .project was preparation for the . . .'

P8L17: '. . . forecasts made it possible to use . . .'

P8L22: a.o. ??

P9L15: 'adds'

P9L28: remove 'also'; '. . .test was updated to cover 2012 as well.'; Make sure the cases in your sentences and paragraphs match. This is not the only occurrence of this.

P9L10: explicitly mention that 2003 was the heatwave

P9L29: '. . . in a new way to investigate behavior. . .'

P9L31: something seems to be missing in this sentence. As compared to what??

P11L20: was aimed at?? Is still primarily aimed at??

P11L30: remove 'the'

P12L5: '. . .is not currently taken. . .'

P12L11: replace 'also' with 'and'

P12L13: 'The default for the model is 3-hourly ECMWF. . .'

P12L15: 'online' is one word

P12L22: do you mean 'variables' when you use 'relations'? or relationships? Relations is not correct.

P12L25: 'are' not 'was'

P13L8: 'in the vertical, fixed emission profiles. . .'

P13L9: 'If desired, scenario factors for specific . . .. can be integrated without changes to the code.'

P13L19: 1/112th as a decimal for consistency.

P13L23-24: '. . .database, were relabeled as "ocean or sea", since. . .'

P13L27: replace 'over' with 'between'; '. . .and below-cloud scavenging.'

P14L13-14: move 'also' to '. . . data are also available. . .'

P14L22-23: move 'often' to '. . .models are often used. . .'

Section 5.2: consistency in case – sometimes past, sometimes present, edit this

P15L7: '.. all data points with . . .'

P15L14: write out RMSE the first time it is used

P15L31: 'Figure 4 illustrates a time series for ozone and PM10 for a station in the Netherlands, comparing observed and modeled concentrations.'

-There are a lot of other edits that should be taken care of for English, please have a native English speaker read through the paper to catch these things.

P19L31/P20L10: why are these words bolded?

*These are bolded to guide the eye of the reader to the species that is treated in the paragraph*

P20L17: the BL is part of the troposphere, please edit wording to make this clear.

P20L16: include a reference for the original nucleation scheme by Vehkamäki

P21L16: 'the role of reactive nitrogen becomes more clear' please rephrase. How so?
P21L22-23: how are met conditions going to be modeled more realistically in the future?
Do the authors mean that 'by taking met conditions into account', the soil Nox,
etc will be modeled more accurately?
Table 1: for anthro emissions, please be consistent with the EI abbreviation. Previously
it was TNO-MACC-III; for fire emission, is this the same as the Kaiser ref? please add
it if so.
Figure 4: please add labels to each of the plots (a, b, c) and update the references to
them in the text. In the legend, an explanation for the red dots is missing.

---

## Author Comment (AC2) · 5 Sep 2017

*We thank Dr. Rouil for her valuable comment, that contributed to the quality of the revised paper. The response to her comments is below in Italics.*

The idea of a curriculum vitae for the LOTOS-EUROS model, which is one of the mostfamous and efficient chemistry-transport model in Europe is very interesting and welcome in this period when, as mentioned by the authors such models have reached a certain level of maturity. The article also introduces some history about the development of air quality models over the twenty past years which is also well documented. As a general comment, we have here a good and relevant paper. The main added-value of this paper is to explain the history of a chemistry-transport model like LOTOS-EUROS, the drivers that led to the current open-source version, the current and future challenges. In that perspective it would have been interesting to get more details about the merging process between both initial models LOTOS and EUROS. EUROS is briefly described compared to LOTOS and how the Dutch teams managed to get only one model is not really explained although this is a quite original approach. For instance, it would be relevant to know how the experts selected the model parametrizations, etc .,,(in paragraph 2.1).

*This is a good point. The merging was quite a pragmatic process in which LOTOS was used as a backbone for the structure. Many of the parameterizations were quite comparable between the models, and where EUROS had a more advanced scheme this was included (in particular POP and deposition). A brief remark was added to the text.*

I understand the authors decided to tell a story avoiding equations describing more in detail the model (this is not a peer-reviewed model description). This choice is valuable to focus on the model development strategy, but as a consequence, some parts of the paper may remain a little unclear for non experts readers (reference to EnKF method and VBS model in paragraphs 6.1 and 6.4 for instance or the paragraph on emissions modelling -6.3- which may be confusing for someone who did not understand that emissions modelling is a part of the CTM).
*Brief explantations on VBS, EnKF and emission modelling were added to the text*

Also, the discussion on high resolution runs (page 22) is relevant but not conclusive (finally is the combination of LOTOSEUROS and OPS the most promising approach? addedd avlue of the plume-in-grid?) I would recommend to review those paragraphs and to amend them with few more explanations.
*The present OPS-LOTOS-EUROS coupling is a useful product for specific applications, but does not fully expoit the possibilites of a more advanced plume in grid approach. This is added tot he text.*

This paper is pleasant to read for air quality modellers who share the authors' concerns and philosophy but may be more difficult for people outside the field. It is mainly dueto the fact that a number of references and definitions, useful for good understanding, is missing . I have noted the following terms used without any explanations: GEOSS, PoDY, Ensemble approach, NOy, SDS-WAS and some references to models : HARMONIE, COSMO, VBS, OPS... few words to introduce them are necessary
*Clarifiactions were added for GEOSS,PoDY,  SDS-WAS, VBS, HARMONIE and COSMO, OPS*
*The meaning of ensemble should be clear from the context.*

Please not that schemes provided on figure 2 are very difficult to understand, and for this reason are almost useless. It is necessary to revise and simplify them, making the acronyms more explicit and focusing on the key messages those schemes are supposed to bear. The other figures are correctly chosen.

*We agree that Figure 2 may contain too much detail for the general reader. It was drastically simplified, thereby better highlighting the natural cooperation between RIVM, KNMI and TNO as model consortium partners. In addition the aspect of data assimilation is now illustrated more clearly.*

I would just recommend to add, if possible, a representation of modelling fields or their performances with assimilationof satellite information. This aspect is well discussed in the paper but not illustrated.

*We feel that Figure 6 demonstrates the impact of data assimilation (albeit with ground observations) quite well and its caption has been extended to give more information. Adding another figure on data assimilation would give the topic too much weight for this paper.*

A good point is the extensive bibliography provided. However, I would recommend to add a reference (for example page 4 line 15 with Menut et al. 2013) to the last peerreviewed paper related to the evolutions of the CHIMERE model (https://www.geoscimodel- dev.net/10/2397/2017/).
*The reference is added*

Here are a number of typo corrections I noted: - Page3 line 16 a parenthesis is missing after 1972; - Page 16 line 31 "months" instead of "mont", - Page 21 line 26 a dot is
missing - Page 22 line 17, add a "s" to "application", - Page 22 line 18 "from" instead
of "form", line 19 add a reference for HARMONIE and COSMO (at least institutes that
develop them), - A number of times in the text PM10, PM2,5, NO2 or NOX are written
without indices - Check the References to "Collette" , the correct name is "Colette"
*These and several other minor textual corrections were made*